# AMBER: Adaptive Mesh Generation by Iterative Mesh Resolution Prediction

**Niklas Freymuth**[1][*] **Tobias Würth**[2] **Nicolas Schreiber**[1] **Balazs Gyenes**[1] **Andreas Boltres**[1][3]
**Johannes Mitsch**[2] **Aleksandar Taranovic**[1] **Tai Hoang**[1] **Philipp Dahlinger**[1]
**Philipp Becker**[1] **Luise Kärger**[2] **Gerhard Neumann**[1]

[1]Autonomous Learning Robots, Karlsruhe Institute of Technology, Karlsruhe
[2]Institute of Vehicle System Technology, Karlsruhe Institute of Technology, Karlsruhe
[3]SAP SE

## Abstract

The cost and accuracy of simulating complex physical systems using the Finite Element Method (FEM) scales with the resolution of the underlying mesh. Adaptive meshes improve computational efficiency by refining resolution in critical regions, but typically require task-specific heuristics or cumbersome manual design by a human expert. We propose Adaptive Meshing By Expert Reconstruction (AMBER), a supervised learning approach to mesh adaptation. Starting from a coarse mesh, AMBER iteratively predicts the sizing field, i.e., a function mapping from the geometry to the local element size of the target mesh, and uses this prediction to produce a new intermediate mesh using an out-of-the-box mesh generator. This process is enabled through a hierarchical graph neural network, and relies on data augmentation by automatically projecting expert labels onto AMBER-generated data during training. We evaluate AMBER on 2D and 3D datasets, including classical physics problems, mechanical components, and real-world industrial designs with human expert meshes. AMBER generalizes to unseen geometries and consistently outperforms multiple recent baselines, including ones using Graph and Convolutional Neural Networks, and Reinforcement Learning-based approaches.

## 1 Introduction

Physical simulations are a fundamental tool in a wide range of science and engineering applications. As simulations become more complex, researchers and practitioners increasingly rely on numerical solutions to intricate Partial Differential Equations (PDEs). The Finite Element Method (FEM) discretizes complex geometries into simpler mesh elements and solves the resulting system of linear equations [1–4]. The FEM is ubiquitous in numerical engineering, finding application in fluid flow simulations [5], structural mechanics [6, 7], electromagnetics [8], and injection molding [9].

For such simulations, both the simulation cost and accuracy scale with mesh resolution. Therefore, adaptive meshing, which assigns more mesh elements to key regions of the geometry, is essential for efficient and accurate simulations [10, 11]. An example is structural analysis in the automotive industry [7], where FEM is used to model complex components under varying forces and stresses. Figure 1 shows such a component, a car seat crossmember, where a finer mesh is required near bends and holes. Traditional Adaptive Mesh Refinement (AMR) techniques iteratively refine existing meshes using predefined heuristics based on problem geometry and process conditions [12–14]. Similarly, Adaptive Mesh Generation (AMG) generates meshes from functions such as sizing fields, which define local element sizes on the geometry [15, 16]. However, both methods are still limited in efficiency and adaptability to new applications. As a result, adaptive meshing in practice requires

---

[*]correspondence to `niklas.freymuth@kit.edu`

39th Conference on Neural Information Processing Systems (NeurIPS 2025).

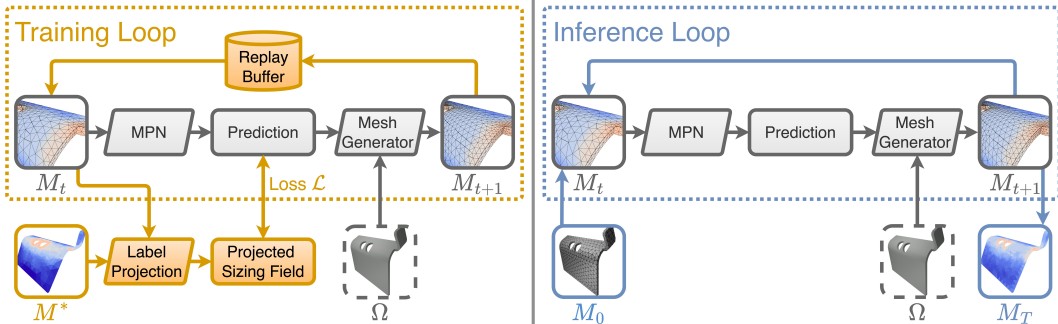

Figure 1: *AMBER* learns adaptive mesh generation on complex geometries for simulation applications from an expert dataset. **Left:** During training, *AMBER* predicts a sizing field, as indicated by the mesh's color, from labels projected from an expert mesh $M^*$. *AMBER* continuously updates a replay buffer with newly generated meshes to preserve a diverse and accurate training data distribution. **Right:** During inference, *AMBER* starts from an initial mesh $M_0$, predicts a sizing field per element, and feeds it into a mesh generator that refines the mesh using the underlying geometry $\Omega$. This process is repeated until a final mesh $M_T$ is produced. On the car seat crossmember shown above, *AMBER* learns that the expert assigns more mesh elements to holes and sharp bends, which are particularly interesting for strength and durability analyses.

significant manual input and domain expertise. Engineers often hand-tune local mesh resolutions for each new geometry or problem [15, 17, 18]. This repetitive and time-consuming process creates bottlenecks in applications like iterative design and process optimization.

To address this issue, we propose Adaptive Meshing By Expert Reconstruction (*AMBER*), a data-driven method for iterative AMG. *AMBER* employs a Message Passing Network (MPN) [19, 20], a class of Graph Neural Networks (GNNs) [21], to predict target element sizes across a sequence of mesh refinement steps. Trained on small datasets, each consisting of roughly 20 geometries and corresponding expert meshes, *AMBER* learns underlying meshing strategies and tackles the core challenge of extreme local variation in element sizes. Unlike prior learned AMG approaches [22–24], *AMBER* iteratively generates meshes using each intermediate mesh's vertices as sampling points to predict the next target sizing field. This iterative scheme, together with the MPN, enables adaptation to non-uniform geometries, while simultaneously adjusting local sampling resolution in response previous mesh generation steps. As a result, *AMBER* is highly effective in adaptive meshing, where spatially varying target sizes necessitate correspondingly localized prediction densities.

Figure 1 shows an overview of our method. At inference time, *AMBER* starts from a coarse initial mesh and iteratively predicts sizing fields to feed into an out-of-the-box mesh generator [25], which generates an adapted mesh. During training, predicted sizing fields are supervised by projecting element sizes from expert meshes onto intermediate meshes. To address the distribution shift introduced by intermediate meshes during inference, we maintain a replay buffer populated with meshes generated by the model itself. This strategy echoes online imitation learning approaches such as DAgger [26], but replaces the human-in-the-loop with automatic data generation and labeling. In doing so, *AMBER* bootstraps [27] its own training distribution, implicitly performing data augmentation [28] by including meshes on different local scales, to stabilize learning and inference.

We evaluate our method on six novel datasets introduced in this work, covering a wide range of 2D and 3D geometries meshed by human experts and heuristics[2]. These geometries vary in difficulty and model a diverse set of common engineering problems. We compare *AMBER* against supervised learning [22, 29, 30] and Reinforcement Learning (RL) [31] baselines. *AMBER* consistently produces higher-quality meshes than all baselines, both in terms of visual quality and quantitative metrics. We additionally explore the runtime of *AMBER*'s components. We find that *AMBER*'s cost is dominated by the final mesh generation step, which is required for all mesh generation methods, and that the full *AMBER* mesh generation process is faster and scales significantly better than classical iterative error estimation methods. Furthermore, we present extensive ablations to show the effects of individual design choices, such as loss, refinement steps, and sizing field parametrization.

---

[2]Project page, code and datasets are available at `https://niklasfreymuth.github.io/AMBER`.

To summarize our contributions, we **(1)** propose *AMBER*, a novel approach for Adaptive Mesh Generation (AMG) that produces a sequence of meshes, using each intermediate mesh to predict a target resolution for the next mesh, **(2)** introduce six new datasets spanning both 2D and 3D geometries, designed to reflect realistic and diverse problem settings; two of these include human-generated meshes, and **(3)** conduct extensive experiments demonstrating that *AMBER* produces significantly better meshes than state-of-the-art supervised and RL methods on these datasets.

## 2 Related Work

**Meshing for Simulation.** Modern meshing approaches either use Adaptive Mesh Refinement (AMR), which refines an existing mesh [10, 32], or Adaptive Mesh Generation (AMG), which generates a new mesh [33–36]. Typical AMR techniques rely on heuristics [12] or error estimates [13, 14], which can be inaccurate, unreliable, or computationally expensive [14, 37, 38]. In contrast, Adaptive Mesh Generation (AMG) methods generate new meshes from geometric or solution-derived features over the domain, such as curvature or Hessian information, to prescribe local element size and potentially anisotropy [39, 40, 16]. While effective in practice [33, 11], they share the shortcomings of AMR and also require task-specific metrics or a tediously hand-crafted target sizing field for each domain [41, 11]. In contrast, we aim to learn scalar sizing fields directly from expert meshes.

**Learning-Based Mesh Generation.** Existing learning based AMG approaches train surrogate models to either directly predict a sizing field or the local solution error, which is inverted to obtain a sizing field. One line of work encodes the domain using a simple, parameterized representation, which is fed to an Multilayer Perceptron (MLP) that either predicts coordinate-conditioned outputs [29, 23], similar to NeRFs [42], or computes the sizing field on a fixed background mesh [43, 44] or as a set of point sources [45]. Huang et al. [22] discretize the domain into a fixed-resolution image and process it with a CNN to directly predict a sizing field. More recent methods use a Graph Convolutional Network (GCN) to operate on the vertices of a coarse mesh. Of these, *GraphMesh* [24] generalizes to arbitrary polygonal domains and improves over prior GCN-based models [46]. *AMBER* also predicts a sizing field on a discrete mesh, but does so iteratively across a sequence of intermediate meshes. This enables dynamic adaptation of the sizing field across scales, without being restricted to any specific domain representation or discretization, allowing it to produce higher quality meshes.

**Learning-Based Mesh Refinement.** Several recent AMR approaches employ learning for mesh refinement by subdivision, i.e., they train a model to iteratively decide which mesh elements to divide into multiple smaller elements. In this class, supervised methods include learning refinement strategies with recurrent networks [47], optimizing element anisotropy based on error estimates [48], and using hand-crafted features to estimate error for adjoint-based refinement [49, 50]. Alternatively, a recent line of work applies RL to AMR by element subdivision [31, 51–53], employing carefully crafted reward functions to quantify the benefit of each refinement. These reward functions typically require an underlying system of equations and either restrict the maximum mesh resolution [53, 31] or encode a specific, heuristic refinement criterion [52]. Out of these methods, Adaptive Swarm Mesh Refinement (*ASMR*) [31, 51] proposes local, element-wise rewards, improving scaling capability and mesh quality over previous work. *AMBER* further improves over *ASMR's* scalability and mesh quality, while avoiding the complicated reward design and the requirement for a Finite Element Method (FEM) in the loop by using expert meshes. Another class of learning-based AMR methods employs mesh movement [11] for refinement [54–56]. These methods start with a uniform mesh and deform its elements, requiring a fixed starting resolution. In contrast, *AMBER* learns to produce a sequence of sizing fields from a coarse uniform mesh, inducing meshes with different numbers of elements. Other mesh movement based methods focus on highly specific tasks, such as fluid dynamics [57, 58], while *AMBER* is task agnostic.

**Graph Network Simulators.** GNNs [21], particularly MPNs [19, 20], are widely popular for mesh-based surrogate simulation [20, 59–63]. MPNs encompass the function class of several classical PDE solvers [64], making them a popular choice for learning representations on meshes [20, 59, 65]. We similarly use MPNs on meshes, but do not learn a simulator. Instead, we generate application-specific adaptive meshes for efficient and robust FEM-based simulation.

**Online Data Generation.** Imitation learning approaches such as DAgger [26] address distribution shift by iteratively querying expert feedback on model rollouts. Bootstrapping methods like pseudo-labeling [66] and Noisy Student [67] expand the training set using model-generated labels.

Replay buffers [68–70] mitigate covariate shift by combining past and current experiences, while data augmentation [28, 71, 72] introduces synthetic variations to enhance generalization. Unlike these approaches, *AMBER* stores model-generated meshes across resolutions in a replay buffer and automatically projects labels onto them. This process effectively augments training data by providing meshes of different resolutions, improving distributional robustness without requiring external supervision or expert relabeling.

## 3    Method

Our training datasets contain $N$ tuples $\{(\Omega, \mathcal{P}, M^*)\}$, each consisting of a geometry $\Omega \subseteq \mathbb{R}^d$ of dimension $d$, an optional set of process conditions $\mathcal{P}$, and a corresponding expert mesh $M^*$. Each geometry describes a closed physical body in 2D or 3D, which is discretized into simplical elements $M_i^*$ on the subdomain $\Omega_i^* \subset \Omega$ by the mesh. We aim to learn a function that takes a geometry $\Omega$ and process conditions $\mathcal{P}$ from the dataset and generates a mesh $M$ that minimizes a distance metric $d(M, M^*)$ to the corresponding expert mesh $M^*$. We make no further assumptions on the structure of the meshes, and use both heuristically refined and human-generated meshes as expert data.

We factorize mesh generation into two parts. First, a learnable function consumes a geometry $\Omega$, process conditions $\mathcal{P}$ and derived features, and outputs a spatially-varying, scalar-valued sizing field $\Omega \to \mathbb{R}_{>0}$. Second, a non-parametric function $g_{\texttt{msh}} : (\Omega \times (\Omega \to \mathbb{R}_{>0})) \to M$ consumes a geometry and a sizing field and returns a mesh approximately conforming to this sizing field. The sizing field describes the desired average edge length of the generated mesh's elements over the domain. We consider isotropic meshes, i.e., meshes where the elements have a roughly equal aspect ratio. In this case, the local sizing field is directly related to the desired volume of the resulting mesh elements.

**Message Passing Network (MPN).** We instantiate our backbone to predict sizing fields using an MPN [19, 20]. An MPN iteratively updates the latent node and edge features over $L$ message-passing steps. We encode mesh vertices as nodes $\mathcal{V}$ and their neighborhood relations as edges $\mathcal{E} \subseteq \mathcal{V} \times \mathcal{V}$ of a bidirectional graph $\mathcal{G}_{\Omega^t} = \mathcal{G} = (\mathcal{V}, \mathcal{E})$. We assign process condition and domain-dependent vertex features $\mathbf{h}_v$ and edge features $\mathbf{h}_e$. Using learned linear embeddings $\mathbf{h}_v^0 = \mathbf{h}_v \mathbf{M}_v$ and $\mathbf{h}_e^0 = \mathbf{h}_e \mathbf{M}_e$ of the initial node and edge features, each step $l$ computes features

$$\mathbf{h}_e^{l+1} = \mathbf{h}_e^l + \psi_{\mathcal{E}}^l(\mathbf{h}_v^l, \mathbf{h}_u^l, \mathbf{h}_e^l), \text{ with } e = (u, v), \qquad \mathbf{h}_v^{l+1} = \mathbf{h}_v^l + \psi_{\mathcal{V}}^l(\mathbf{h}_v^l, \bigoplus_{e=(v,u) \in \mathcal{E}} \mathbf{h}_e^{l+1}).$$

The permutation-invariant aggregation $\bigoplus$ can be realized via, e.g., a sum, mean, or maximum operator. All $\psi_{\mathcal{E}}^l$ and $\psi_{\mathcal{V}}^l$ are parameterized as learned MLPs. The output of the final layer is a learned representation $\mathbf{h}_v^L$ for each node $v \in \mathcal{V}$. We feed this representation into a decoder MLP to yield a prediction $x_j = \text{MPN}(\mathcal{G}, \mathbf{h}_v, \mathbf{h}_e)_j$ per node $v_j \in \mathcal{V}$, which we abbreviate as $\text{MPN}(v_j)$.

**Mesh Generation.** We refer to the non-parametric function $g_{\texttt{msh}}$ as the *mesh generator*. It creates a mesh that matches the desired sizing field under several criteria on the elements, such as their aspect ratio and size gradation. This results in well-behaved elements and a smooth transition between element sizes. While different mesh generators exist, we use the Frontal Delaunay algorithm implemented in GMSH [25] for simplicity.

### 3.1    Iterative Mesh Generation with *AMBER*

**Predicting a Sizing Field.** Given a geometry $\Omega$ and task-specific process conditions $\mathcal{P}$, a coarse, uniform initial mesh is generated for the initial $M^t$ with $t = 0$. This mesh is then encoded as a graph and processed using an MPN to predict the *discrete* sizing field $\hat{f}(v_j)$ over mesh vertices $v_j$, with $\hat{f}(v_j)$ derived from the network's output $x_j$ through a subsequent transformation.

We could alternatively predict sizing values per mesh element, yielding a piecewise constant sizing field. However, since the MPN operates on an intermediate mesh with a different topology from the target mesh, element-level predictions lack the granularity needed for effective refinement. Instead, *AMBER* predicts sizing field values over mesh vertices and applies the interpolant $\mathcal{I}_M(\hat{f})$ to yield a sizing field that is piecewise linear. This interpolant weights the discrete sizing field at the vertices $v_j$ by the mesh's nodal basis functions $\phi_j$ [3], yielding a *continuous* sizing field. Given a point $\mathbf{z} \in \mathbb{R}^d$

we define the interpolant as

$$\mathcal{I}_M(\hat{f})(\mathbf{z}) = \begin{cases} \sum_{j=1}^{|V|} \hat{f}(v_j)\,\phi_j(\mathbf{z}), & \text{if } \mathbf{z} \in \Omega_i, M_i \in \mathcal{N}(v_j) \text{ for some } j, \\ \hat{f}(v_{j'}), & \text{otherwise, where } j' = \arg\min_j \|\mathbf{z} - \mathbf{p}(v_j)\|, \end{cases} \tag{1}$$

where $\mathbf{p}(v_j) \in \mathbb{R}^d$ and $\mathcal{N}(v_j) \subset M$ are the position and element neighborhood of vertex $v_j$, respectively. The fallback to nearest-neighbor extrapolation ensures that the sizing field is defined across all of $\Omega$, including regions outside the discretized mesh domain.

**Iterative Generation.** At step $t$, the mesh generator consumes the continuous sizing field given by $\mathcal{I}_{M^t}(\hat{f})$ and its underlying geometry $\Omega$. Using the vertices of each mech as the sampling points for the next continuous sizing field and repeating this process over $T$ steps results in a final mesh $M^T$. Intuitively, an accurately predicted intermediate sizing field results in a mesh that is more similar to the expert mesh, and therefore provides better sampling points for the MPN to predict the next sizing field even more accurately. Compared to one-step approaches that predict a sizing field on an image [22] or a single coarse mesh [24], *AMBER* therefore automatically adapts its sampling resolution, allowing it to output arbitrarily complex and highly non-uniform meshes where required. We prove convergence of this process in the one-dimensional case under the assumption of perfect predictions in Appendix B. The right part of Figure 1 visualizes this process.

## 3.2 Training *AMBER*

**Predictions and Targets.** Let $V(M_i)$ be the volume of the $d$-dimensional simplicial element $M_i$ of the target mesh. We define the element-wise sizing field as the average edge length of that element $f_e(M_i) = \left(V(M_i)\frac{d!}{\sqrt{d+1}}\right)^{\frac{1}{d}}$. The union over the element's sizing fields induces a piecewise-constant sizing field. To compute the target value $y_j$ of the discrete sizing field at vertex $v_j$ of an intermediate mesh $M^t$, we evaluate the sizing field of the expert mesh $M^*$ at the vertex position $\mathbf{p}(v_j)$. That is, we assign targets $y_j = f_e(M_i^*)$ where $M_i^* \in M^*$ and $\mathbf{p}(v_j) \in \Omega_i^*$. If a vertex lies outside the expert mesh due to, e.g., discretization of the domain, we project it to the nearest element. We could alternatively obtain target values by interpolating the expert sizing field using Equation 1. However, as we show in our experiments, due to *AMBER*'s iterative process, the local resolution of the expert mesh is sufficient to adequately represent the granularity of the solution everywhere.

We train a single shared MPN to regress the target sizing field of the current mesh generation step using a simple Mean Squared Error (MSE) loss. Since sizing fields are strictly positive, we add a softplus transformation to the network's output. To increase the weight of numerically smaller elements in the loss function, we optimize in the untransformed space. Thus, given a prediction $x_j = \text{MPN}(v_j)$, our loss becomes

$$\mathcal{L} = \frac{1}{|V|} \sum_{j=1}^{|V|} \left(x_j - \text{softplus}^{-1}(y_j)\right)^2. \tag{2}$$

We then recover the discrete predicted sizing field as $\hat{f}(v_j) = \text{softplus}(x_j)$.

**Replay Buffer.** During inference, *AMBER* auto-regressively produces a series of intermediate meshes $M^t$. The initial mesh $M^0$ is coarse and uniform. However, the corresponding expert mesh $M^*$ is generally finer and has highly varied topology. To prevent a distribution shift between the training data and the data seen during inference, we therefore maintain a replay buffer [68, 70] of bootstrapped data containing intermediate meshes that *AMBER* generates during training. The replay buffer is initialized with one uniform coarse mesh per expert mesh. After each training epoch, we sample $k$ meshes from the replay buffer for producing new intermediate meshes. For each, we predict a discrete sizing field, generate a new mesh from the induced continuous sizing field, annotate the vertices with a target sizing field using the expert mesh, and store this new labeled mesh in the buffer. The full training pipeline is shown on the left of Figure 1.

## 3.3 Empirical Improvements

Inspired by common best practices, we propose several algorithmic optimizations to further improve *AMBER*'s applicability and efficiency.

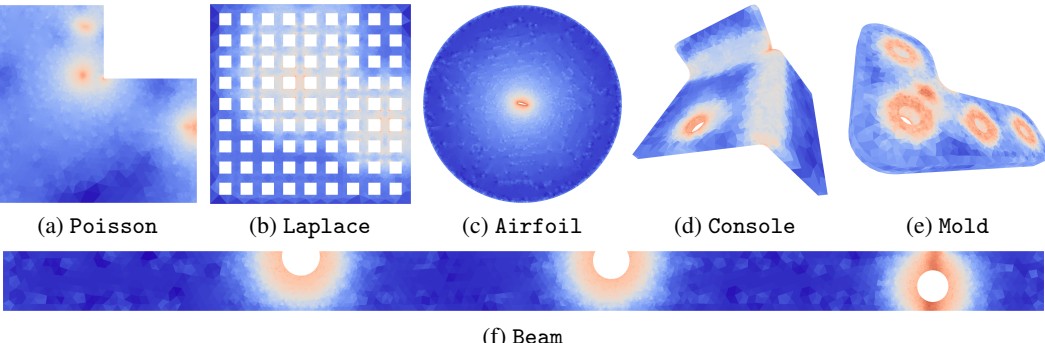

(a) `Poisson`     (b) `Laplace`     (c) `Airfoil`     (d) `Console`     (e) `Mold`

(f) `Beam`

Figure 2: Exemplary *AMBER* meshes for each dataset. The color represents the local element size, with smaller elements being red. We propose six novel and challenging datasets for mesh generation. **(a)** `Poisson` uses an L-shaped domain with a multimodal load function. **(b)** `Laplace` features parameterized 2D lattices with complex Dirichlet boundaries. **(c)** `Airfoil` includes geometries representative of aerodynamic flow setups. **(d)** `Console` consists of 3D car seat crossmembers. **(e)** `Mold` includes complex 3D plates used in injection molding contexts. **(f)**`Beam` covers elongated, perforated beams inducing long-range mesh dependencies.

**Uniform Refinement Depth.** During training, we assign each intermediate mesh a *depth* that corresponds to the number of refinement steps it has undergone from the initial uniform mesh. To reduce distribution shift between inference and training, we enforce a uniform distribution over mesh depths in the replay buffer. When generating new intermediate meshes, we first uniformly sample a target depth and then a mesh with the corresponding depth. We set the maximum depth to the number of refinement steps used during evaluation, $T$.

**Adaptive Batch Size.** Since the meshes in the replay buffer vary greatly in size, using a fixed number of meshes per batch would sometimes lead to out-of-memory errors, or otherwise leave significant available memory unused. Instead, we set a maximum total size over all graphs in a batch, and greedily fill a batch with the least-sampled meshes until it is reached. We define the size of graph as the sum of its number of nodes and edges $s(\mathcal{G}) = |\mathcal{V}| + |\mathcal{E}|$.

**Hierarchical Architecture.** The receptive field of an MPN is determined by the number of message passing steps. As a mesh undergoes iterative refinement, the receptive field can vary significantly across the domain. This makes it challenging to choose appropriate hyperparameters and hinders long-range communication between regions of the graph during the later refinement steps. To ensure a consistent, resolution-invariant receptive field, we employ a hierarchical graph structure that combines the graph $\mathcal{G}^0 = (\mathcal{V}^0, \mathcal{E}^0)$ corresponding to the initial coarse mesh $M^0$ with that of the current interme-diate mesh $M^t$ for all $t > 0$. The hierarchical graph is defined as $\mathcal{G}_{\text{hier}} = \left(\mathcal{V}^0 \cup \mathcal{V}^t, \ \mathcal{E}^0 \cup \mathcal{E}^t \cup \mathcal{E}^{\text{cross}}\right)$, where $\mathcal{E}^{\text{cross}} = \{(v, \pi(v)), \ (\pi(v), v) \mid v \in \mathcal{V}^t\}$ contains bidirectional edges between each interme-diate vertex $v \in \mathcal{V}^t$ and its closest vertex in the coarse mesh $\pi(v) = \arg\min_{u \in \mathcal{V}^0} \|\mathbf{p}(v) - \mathbf{p}(u)\|_2$. We provide a binary node feature indicating the current mesh $M^t$, and mask all node-level features of the initial mesh, using it solely to provide consistent topological connectivity.

**Input/Output Normalization.** We normalize all network inputs, i.e., all node and edge features, to have zero mean and unit variance. The labels are normalized similarly, and the inverse normalization is applied to map predictions back to the original scale. Since the data distribution evolves as new meshes are added to the replay buffer, we maintain running statistics for each input and target feature.

**Residual Prediction.** We improve training stability by predicting the residual between the target sizing field $y_j = f_e(M_i^*)$ and the current discrete sizing field $b_j = f(v_j)$. Given the element neighborhood $\mathcal{N}(v_j)$ of vertex $v_j$ and element-based sizing fields $f_e(M_i)$, we compute the current discrete sizing field $b_j$ at $v_j$ from the current mesh as the convex combination

$$b_j = f(v_j) = \sum_{M_i \in \mathcal{N}(v_j)} \frac{V(M_i)}{\sum_{M_i \in \mathcal{N}(v_j)} V(M_i)} f_e(M_i). \tag{3}$$

We now recover the predicted sizing fields by $\hat{f}(v_j) = \text{softplus}(x_j + \text{softplus}^{-1}(b_j))$, and adapt the loss in Equation 2 accordingly.

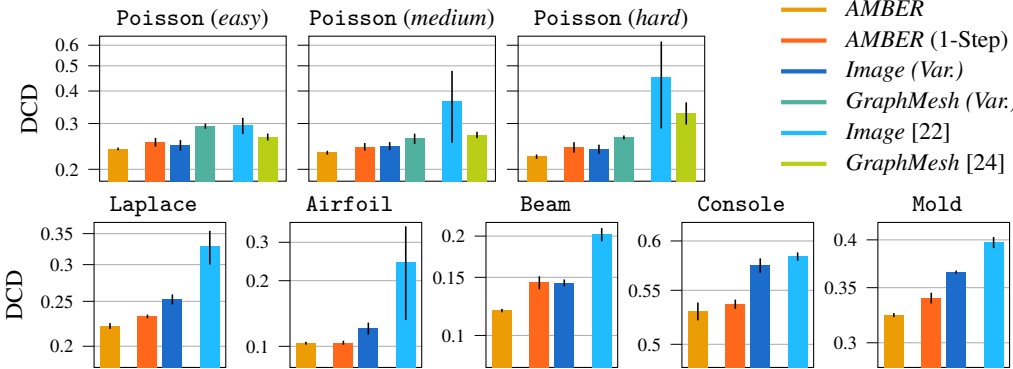

Figure 3: Mean and two times standard error of expert mesh similarity evaluated by Density-Aware Chamfer Distance (DCD) (lower is better). *AMBER* achieves the best results across all datasets, demonstrating its ability to generate highly accurate meshes on diverse and challenging domains. All methods perform well on `Poisson` (*easy*). As task complexity increases, the baselines and eventually variants become less reliable. *AMBER (1-Step)* remains strong across tasks, while the full model achieves further improvements through iterative refinement.

**Scaling Sizing Fields.** We can scale the resolution of generated meshes by introducing a simple refinement constant $c_t$ depending on the generation step $t$, such that the next generated mesh is $M^{t+1} = g_{\mathtt{msh}}(\Omega, c_t \mathcal{I}_{M^t}(\hat{f})(\mathbf{z}))$. While the predicted intermediate meshes allow *AMBER* to adaptively refine its sampling resolution, reaching the full resolution of the expert mesh is unnecessary and computationally expensive. To mitigate this, we set $c_t > 1$ for $t < T - 1$ to coarsen intermediate meshes, starting at the first step $t=0$. Here, setting an exponentially decaying $c_t$ reduces the number of elements for intermediate meshes without reducing the accuracy of the final mesh $M^T$. Additionally, during inference, we can also set $c_{T-1} < 1$ to generate meshes that have a higher resolution than the expert. This adaptation allows the model to flexibly adapt to a given element budget without retraining, which enables zero-shot generalization through a single scalar parameter.

## 4 Experiments

**Datasets and Features.** We introduce six novel datasets representing realistic FEM problems that need adaptive meshing to meet common efficiency and accuracy requirements. The datasets span 2D and 3D domains, as well as diverse applications in physics-based simulation, structural mechanics, and industrial design. Depending on the dataset, we generate geometries procedurally, or source them from openly available or custom datasets. For datasets without a concrete underlying system of equations, we generate meshes from human experts and manually designed, specialized heuristics. Other datasets consider a concrete problem, where we employ an iterative refinement heuristic that utilizes a FEM error indicator. Using this heuristic, we create *easy*, *medium*, and *hard* variants of the `Poisson` dataset to provide expert meshes on different scales. Here, more refinement steps results in an expert mesh with more elements and a larger difference between the largest and smallest elements increases, making the dataset more challenging. Figure 2 illustrates representative *AMBER* meshes from the test set of each dataset. Across datasets, mesh resolution ranges from 1 042 to 65 191 elements.

Appendix D provides training details for *AMBER* and Appendix E details the mesh generation process. We derive dataset-specific features as a function of the process conditions $\mathcal{P}$ for `Poisson`, `Laplace` and `Mold`, as detailed in Appendix C. The `Poisson` and `Laplace` datasets use a FEM solver in the loop for expert mesh generation via an iterative refinement heuristic. For these datasets, we therefore provide FEM solutions as a vertex-level input feature for each mesh. For all datasets, we add several geometric features, as detailed in Appendix D.3.

**Evaluation.** We evaluate the generated meshes by comparing their local resolution to that of an unseen expert reference mesh on the same geometry and process conditions, using five random seeds per experiment. First, we use the Density-Aware Chamfer Distance (DCD) [73] over both mesh's vertices. The DCD is a symmetric, exponentiated variant of the Chamfer distance that allows multiple

points in one set to match a single point in the other. Semantically, it treats both vertex sets as samples from an unknown density. Second, we compute a symmetric relative projected $L^2$ error between the sizing fields induced by the evaluated and expert meshes. In contrast to the DCD, this metric captures discrepancies in local element sizes. The combination of these two metrics with different semantic interpretations is robust against potential artifacts in the generated meshes. Finally, we evaluate downstream simulation quality versus number of mesh elements for the `Poisson` task by using the norm of the error indicator of Equation 10. Appendix F details all metrics.

**Baselines and Variants.** We compare to *GraphMesh* [24], which is based on a two-step GCN. *GraphMesh* relies on mean value coordinates [74] and is limited to polygonal domains, only allowing us to evaluate it on the `Poisson` dataset. *Image* [22] predicts either pixel- or voxel-wise sizing fields from binary geometry masks of a discretized domain using a 2D or 3D Convolutional Neural Network (CNN), respectively. We adapt both baselines to use softplus-transformed predictions. This transformation is omitted in the original works, which focus on relatively simple problems where training instabilities are less pronounced. Without it, models tend to diverge, producing overly fine meshes. To disentangle training and algorithmic design, we additionally introduce *Variants* of each baseline that incorporate our loss (Equation 2) and normalization. Additionally, we compare against *AMBER (1-Step)*. This variant runs a single *AMBER* generation step by setting $T=1$, which demonstrates the benefit of iterative mesh generation.

We compare against Adaptive Swarm Mesh Refinement (*ASMR*) [51] as an RL baseline. *ASMR* learns a policy to iteratively mark elements for AMR, optimizing a reward function tied to the improvement of a specific FEM solution. This reward requires a fine-grained uniform reference mesh to compare to, whose resolution bounds the maximum number of refinements. In Appendix H.9, we also explore a variant that omits the reference mesh in favor of using the error indicator from Equation 10. This modification enables deeper refinement and mesh resolutions comparable to those of the expert. We evaluate *ASMR* on the `Poisson` task. Appendix G details all baselines and variants.

**Runtime and Cross-Dataset Generalization.** We measure the runtime of *AMBER* and its individual components for `Poisson` (*easy/hard*) and compare it to the expert heuristic used to generate the data. We additionally explore *AMBER*'s ability to generalize across datasets by training a single model on joint data of `Poisson` (*hard*), `Laplace` and `Airfoil`. We concatenate the 20 expert meshes per task into 60 total training meshes, using a shared replay buffer for the data. We one-hot encode the task in the node features, and zero out task-specific features when unavailable. We do not change any other training or inference hyperparameters. We call this variant *AMBER* (Mixed).

**Additional Experiments.** For *AMBER*, we explore the loss in Equation 2 and the components from Section 3.3. We also vary training data size and test alternative sizing field parameterizations for $\hat{f}$. For the *Image (Variant)* baseline, we explore lower input resolutions and versions that omit either the loss or input/output normalization. Appendix H provides additional details.

## 5 Results

**Quantitative Results.** Figure 3 evaluates Density-Aware Chamfer Distance (DCD) over vertex sets to the expert mesh across datasets. On `Poisson` (*easy*), all methods perform well. As complexity increases for, e.g., `Poisson` (*medium/hard*), our training procedure shows more significant benefits, causing both variants to significantly outperform their published baselines. Across datasets, the *AMBER (1-Step)* produces accurate sizing field predictions and high-quality meshes closely matching the expert. It also generalizes to 3D, where the *Image* methods struggle. *AMBER* further improves mesh quality, likely due to its iterative mesh generation. Here, multiple generation steps allow the intermediate meshes, which govern the prediction resolution, to adapt dynamically to the underlying geometry, improving mesh quality in complex regions. Appendix H.1 shows consistent trends using a symmetric $L^2$ error, supporting *AMBER*'s ability to generate high-quality meshes. Appendix H.2 matches these results on `Poisson` (*hard*) and `Laplace` for the norm of the error indicator of Equation 10, which requires a concrete system of equations to evaluate. This strong correlation between the error indicator and DCD across methods supports the use of DCD as a reliable proxy on datasets where downstream simulation error is not directly available.

Figure 4 compares *AMBER*, *ASMR*, and the expert meshes using the per-mesh norm of the same indicator for `Poisson` (*easy/medium/hard*). We obtain Pareto fronts by varying *ASMR*'s element penalty and scaling *AMBER*'s predicted sizing field at inference between 0.5 and 2.0. All methods

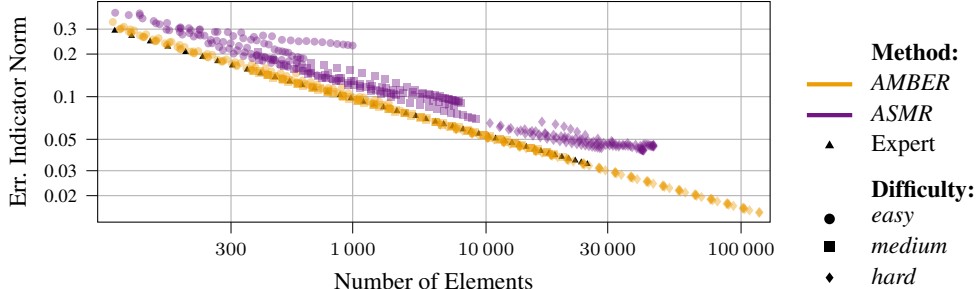

Figure 4: Log-log plot of error indicator norm versus number of mesh elements (lower left is better) for *AMBER*, *ASMR* and the expert across `Poisson` (*easy*, *medium*, *hard*). Each marker shows the mean over the test set for a given seed. *AMBER* and *ASMR* evaluations are obtained by scaling the final predicted sizing field and tuning the element penalty, respectively. *AMBER* closely matches or even exceeds expert performance in terms of indicator error, and generalizes to meshes that are more than $3\times$ finer, maintaining the expected error-element trend beyond $100\,000$ elements.

Table 1: Generalization across datasets. Comparison between *AMBER* trained individually per dataset and *AMBER* (Mixed) trained jointly on all datasets. The mixed model achieves nearly identical performance, indicating strong generalization and potential for multi-task learned mesh generation.

| **Method** | Poisson (*hard*) | Laplace | Airfoil |
|---|---|---|---|
| *AMBER* | $0.224 \pm 0.004$ | $0.222 \pm 0.003$ | $0.103 \pm 0.002$ |
| *AMBER* (Mixed) | $0.226 \pm 0.011$ | $0.222 \pm 0.005$ | $0.102 \pm 0.002$ |

follow the expected log-log error–element trend [75]. Markers show test-set averages per target resolution and random seed. *ASMR* exhibits high variance across seeds and degrades beyond $\sim 30\,000$ elements due to its fixed-depth reference mesh. In contrast, *AMBER* closely matches and slightly surpasses expert performance on fine meshes, likely due to smoother mesh generation. It also generalizes to $>100\,000$ elements, even though the largest expert mesh has only $31\,510$ elements. This generalization only requires adjusting a single scalar, enabling zero-shot, budget-aware mesh generation without retraining.

**Runtime and Cross-Dataset Generalization.** Appendix H.3 compares *AMBER*'s runtime with that of the expert heuristic used to generate `Poisson` data. For the same number of elements, both methods achieve a similar error indicator norm. Yet *AMBER* is significantly faster on finer meshes, outperforming the iterative expert heuristic by more than an order of magnitude on meshes with more than $30\,000$ elements. We additionally find in Table 6 that *AMBER*'s runtime is dominated by its last mesh generation step, which is needed for any mesh generation method.

Table 1 explores *AMBER*'s ability to train on multiple datasets at the same time. *AMBER* (Mixed) shows the approximately equal performance to *AMBER* on all considered tasks, opening up interesting avenues for multi-task and general-purpose learned mesh generation algorithms in future work.

**Qualitative Results.** Figure 2 shows a final *AMBER* mesh per dataset and Figure 5 shows a close-up of generated meshes for different supervised methods on `Console`. Both figures show that *AMBER* produces accurate sizing fields on diverse domains and geometries and produces high-quality meshes, closely resembling the expert. In contrast, the *Image* baselines only learn general, low-frequency features of the expert's sizing field, but fail to capture finer details. The result is a comparatively more uniform, less adaptive mesh. Figure 6 provides a full *AMBER* rollout on `Console`, showcasing the iterative generation process. In each step, *AMBER* consumes the previous mesh, using it to predict an increasingly accurate sizing field. We provide further visualizations for *AMBER* rollouts and generated meshes for all baselines in Appendix I.

**Additional Experiments.** Appendix H.4 validates the algorithmic improvements from Section 3.3. In particular, the hierarchical architecture, the loss and normalization are critical for performance. Appendix H.5 finds piecewise-linear sizing fields work better than piecewise-constant ones. Appendix H.6 shows that *AMBER* benefits modestly from additional data, and generalizes well from

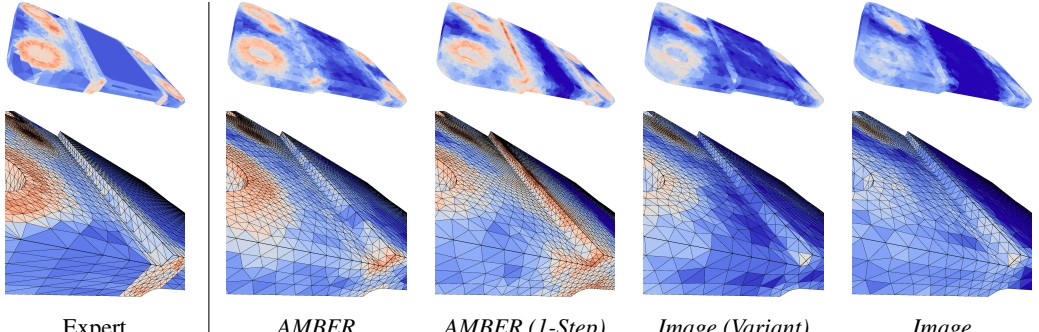

| Expert | | *AMBER* | *AMBER (1-Step)* | *Image (Variant)* | *Image* |

Figure 5: Full views and close-ups of generated `Mold` test meshes. The element size is denoted by color, with red indicating small elements. *AMBER* closely matches the expert mesh, producing finer elements near the hole and coarser elements near the mesh's border. In comparison, the *Image* baselines have less variation in the element size, matching the expert less closely.

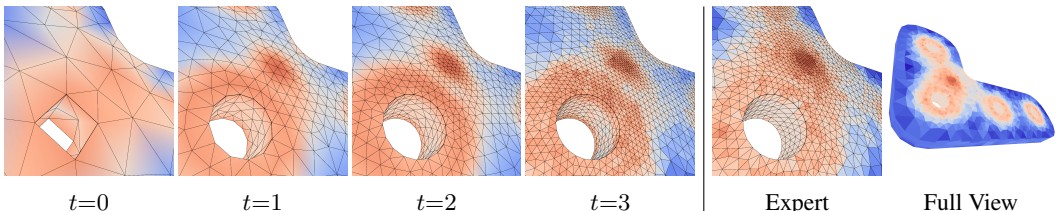

| $t=0$ | $t=1$ | $t=2$ | $t=3$ | Expert | Full View |

Figure 6: Close-ups of intermediate and final *AMBER* meshes on `Mold`, contrasted with the expert mesh. The color for intermediate meshes denotes the predicted sizing field (red is small), which is given to a mesh generator to produce the next mesh. The final mesh's color denotes its element size.

only five training samples on several datasets. This data-efficiency likely stems from *AMBER*'s Euclidean-invariant MPN architecture and implicit data augmentation. We find in Appendix H.7 that *AMBER* improves for more mesh generation steps, converging at around three steps. Appendix H.8 explores different configurations of *Image (Variant)*, showing the importance of image resolution, loss function, and normalization. Finally, Appendix H.9 introduces an *ASMR* variant with the error indicator as reward. While this version avoids *ASMR*'s degradation on fine meshes, it performs worse overall, likely due to a weaker reward signal.

## 6   Conclusion

We introduce *AMBER*, a novel method for iterative Adaptive Mesh Generation (AMG) that combines a replay buffer of bootstrapped data with Message Passing Graph Neural Networks operating on intermediate meshes. At each step, *AMBER* consumes the current mesh to predict a target resolution for the next one, allowing fine-grained adaptation to complex geometries. *AMBER* generates high-quality adaptive meshes across six novel datasets spanning diverse and realistic 2D and 3D geometries, consistently outperforming supervised and reinforcement learning baselines. These results demonstrate the effectiveness of learning-based approaches in reducing manual meshing effort, contributing toward more efficient and scalable simulation workflows in engineering applications. Appendix A discusses the broader impact of our work.

**Limitations and Future Work.** *AMBER* predicts scalar-valued, piecewise-linear sizing fields, which limits expressiveness in scenarios requiring extreme variation in local mesh density, or anisotropic refinement. Predicting tensor-valued sizing fields or using higher-order polynomials is a promising direction for future work. Furthermore, our experiments indicate that the same model can be trained on different datasets, showing that there is potential to train a general-purpose model across a vast amount of datasets. Lastly, assessing the performance of *AMBER* directly through simulation error metrics on real-world scenarios is an interesting avenue for future research.

## Acknowledgements

This work is part of the DFG AI Resarch Unit 5339 regarding the combination of physics-based simulation with AI-based methodologies for the fast maturation of manufacturing processes. The financial support by German Research Foundation (DFG, Deutsche Forschungsgemeinschaft) is gratefully acknowledged. This work is additionally funded by the German Research Foundation (DFG, German Research Foundation) - SFB-1574 – 471687386. The authors acknowledge support by the state of Baden-Württemberg through bwHPC, as well as the HoreKa supercomputer funded by the Ministry of Science, Research and the Arts Baden-Württemberg and by the German Federal Ministry of Education and Research.

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

# A   Broader Impact

The proposed method, Adaptive Meshing By Expert Reconstruction (*AMBER*), has the potential to benefit numerous domains that depend on computational modeling and simulation by significantly reducing computational costs without compromising accuracy. This efficiency can expand the scope of feasible simulations in engineering design, and support the deployment of simulation-based tools in resource-constrained environments. Nonetheless, as common with advanced computational tools, there is a risk of misuse in contexts such as weapons development or unsustainable resource exploitation.

# B   Theoretical Convergence of the Iterative Mesh Generation Process

In this section, we provide a convergence proof of the iterative mesh generation process of *AMBER* in a simplified one-dimensional setting. We consider the unit interval as the domain of interest:

$$\Omega = [0, 1] \subset \mathbb{R}.$$

A one-dimensional mesh $M$ is defined as a set of points

$$M = \{v_1, \ldots, v_N\}$$

such that $v_1 = 0$, $v_N = 1$, and $v_i < v_{i+1}$ for all $i = 1, \ldots, N-1$. The sizing field $f_e(M)$ induced by the mesh is directly related to the spacing between points and is defined for $z$ such that $v_i \leq z < v_{i+1}$[3] as

$$f_e(M)(z) := v_{i+1} - v_i, \tag{4}$$

which is defined for the general setting in Section 3.2.

We construct a mesh generator $g_{\mathrm{msh}}$ that, given a sizing field $f : [0,1] \to \mathbb{R}_{>0}$, generates a mesh as follows: set $v_1 := 0$, and define

$$v_{i+1} := \min(v_i + f(v_i), 1).$$

We terminate the process when $v_{i+1} = 1$, resulting in a mesh $g_{\mathrm{msh}}(f) = \{v_1, \ldots, v_N\}$. It is easy to see that

$$v_{i+1} = \sum_{j=1}^{i} f(v_j), \tag{5}$$

for intermediate points $i < N - 1$. Note that this generator acts as an inverse to the sizing field in the sense that

$$g_{\mathrm{msh}}(f_e(M)) = M.$$

Given a mesh $M^t = \{v_1^t, \ldots, v_N^t\}$ and a target mesh $M^*$, we assume perfect predictions and define an interpolated sizing field as

$$\mathcal{I}_{M^t}(\hat{f})(z) := (1 - d)\, f_e(M^*)(v_i^t) + d\, f_e(M^*)(v_{i+1}^t), \tag{6}$$

for $z = (1 - d)\, v_i^t + d\, v_{i+1}^t$ with $0 \leq d \leq 1$.

Under these assumptions, we can prove the following:

**Theorem 1.** *Let $M^1 = \{v_1^1, \ldots v_{N_1}^1\}$ be an initial mesh and $M^* = \{v_1^*, \ldots, v_N^*\}$ a target mesh. For a given mesh $M^t$, define one iteration of AMBER by*

$$M^{t+1} = g_{msh}(\mathcal{I}_{M^t}(\hat{f})). \tag{7}$$

*Then, it holds that $M^N = M^*$.*

*Proof.* We prove this by induction showing that the first $k$ vertices of the k-th output $\{v_1^k, \ldots v_k^k\} \subset M^k$ are equal to the target vertices $\{v_1^*, \ldots v_k^*\} \subset M^*$.

---

[3]For $z = v_N = 1$, we set $f_e(M)(z) = v_N - v_{N-1}$.

Table 2: Overview of dataset characteristics.

| Name | Dim. | Application | Geometries | Online FEM | Expert Meshes | Process Conditions |
|------|------|-------------|------------|------------|---------------|--------------------|
| Poisson | 2D | Electrostatics | Procedurally generated | Yes | Error indicator | Load function |
| Laplace | 2D | Heat or fluid flow | Procedurally generated | Yes | Error indicator | Dirichlet boundary |
| Airfoil | 2D | Fluid dynamics | Open-source dataset | No | Task-specific heuristic | None |
| Beam | 2D | Mechanical load | Procedurally generated | No | Task-specific heuristic | None |
| Console | 3D | Durability analysis | Closed-source dataset | No | Human labeled | None |
| Mold | 3D | Injection molding | Open-source dataset | No | Human labeled | Inlet position |

The case for $k = 1$ is trivial. Consider now the $k + 1$-th *AMBER* step. It holds for $i \leq k$:

$$\mathcal{I}_{M^k}(\hat{f})(v_i^k) = f_e(M^*)(v_i^k) = f_e(M^*)(v_i^*) = v_{i+1}^* - v_i^*, \tag{8}$$

using Eq. 6 for the first equality, the induction proposition for the second equality, and Eq. 4 for the last equality. From Eq. 5 and using the result from above, we get

$$v_{i+1}^{k+1} = \sum_{j=1}^{i} \mathcal{I}_{M^k}(\hat{f})(v_j^k) = \sum_{j=1}^{i} v_{j+1}^* - v_j^* = v_{i+1}^*, \tag{9}$$

for all $i \leq k$ which proves the desired result. $\square$

## C  Datasets

We propose a total of six novel and varied datasets. Poisson features L-shaped domains with a Gaussian Mixture Model as the load function and zero Dirichlet boundaries, adapted from *ASMR* [31]. We vary the resolution of the expert mesh to define *easy*, *medium*, and *hard* variants. Laplace contains parameterized 2D lattices governed by the Laplace equation with complex Dirichlet boundary conditions, representative of structures used in, e.g., materials design. Airfoil includes flow simulations around airfoil-like shapes, as commonly encountered in aerodynamic engineering. Beam captures elasticity problems in mechanical engineering, using elongated beams with internal circular holes. The elongated beams induce long-range dependencies across the mesh. Console consists of 3D car seat crossmember geometries, parameterized and meshed by a human expert. The resulting meshes are optimized for downstream strength and durability analyses. Mold represents injection molding setups with complex 3D plates, varying inlet positions, and handcrafted expert meshes.

Table 2 summarizes dataset metadata. Poisson and Laplace solve a concrete system of equations to yield a FEM solution over the mesh. This solution is used for expert mesh creation, and as features for the graph that we input into the MPN. For the Mold task, the process conditions $\mathcal{P}$ of each data point are comprised of the inlet position for the molding process, which always lies on the surface of the geometry. As such, we re-use each Mold geometry multiple times with different inlet positions, and generate a suitable expert mesh for each of them.

Table 3 provides detailed statistics on mesh resolution for the training sets. Meshes range from 705 to 116 704 elements in the training data. The 3D datasets, i.e., Console and Mold, have a higher ratio of elements to vertices, as they use tetrahedral instead of triangular elements.

The sections below describe the construction of each dataset, including geometry generation and expert mesh creation. We implement the FEM Poisson and Laplace in SCIKIT-FEM [76]. For these datasets, we generate separate training data for each seed during training, but evaluate on a fixed set of validation and test data points. For the other datasets, we created a fixed set of training, validation and testing data points.

### C.1  Poisson

We consider adaptive, problem-specific meshes for Poisson's equation with zero Dirichlet boundary conditions, given in weak form as

$$\int_{\Omega} \nabla u \cdot \nabla v \, d\boldsymbol{x} = \int_{\Omega} qv \, d\boldsymbol{x} \quad \forall v \in V.$$

Table 3: Number of data points per split and min/mean/max number of vertices and elements per mesh in the training data. †For `Mold`, each geometry is paired with multiple inlet positions and corresponding expert meshes. Each of the $18$ training geometries is used with $3$ inlet positions. We reserve $5$ `Mold` geometries for validation and test, using $1$ and $2$ inlet positions, respectively.

| | # Data Points | | | # Vertices | | | # Elements | | |
|---|---|---|---|---|---|---|---|---|---|
| **Name** | Train | Val | Test | Min | Mean | Max | Min | Mean | Max |
| `Poisson` (*easy*) | 20 | 20 | 20 | 387 | 549 | 674 | 705 | 1 042 | 1 292 |
| `Poisson` (*medium*) | 20 | 20 | 20 | 1 562 | 2 234 | 2 736 | 2 985 | 4 358 | 5 365 |
| `Poisson` (*hard*) | 20 | 20 | 20 | 9 951 | 13 224 | 15 884 | 19 563 | 26 185 | 31 510 |
| `Laplace` | 20 | 20 | 20 | 10 161 | 13 840 | 18 193 | 19 308 | 26 341 | 34 414 |
| `Airfoil` | 20 | 5 | 5 | 20 229 | 20 942 | 22 152 | 39 995 | 41 425 | 43 842 |
| `Beam` | 20 | 10 | 20 | 13 011 | 27 727 | 42 306 | 25 161 | 53 804 | 82 521 |
| `Console` | 19 | 2 | 5 | 2 222 | 6 606 | 10 130 | 7 800 | 25 769 | 41 856 |
| `Mold`† | $3 \times 18$ | $1 \times 5$ | $2 \times 5$ | 7 369 | 13 208 | 22 871 | 33 308 | 65 191 | 116 704 |

Each domain is a randomly generated L-shaped geometry of the form $\Omega = (0,1)^2 \setminus \left( [p_0^{(1)}, 1] \times [p_0^{(2)}, 1] \right)$, with $p_0 = (p_0^{(1)}, p_0^{(2)})$ sampled from $U(0.2, 0.8)^2$. The load function $q\colon \Omega \to \mathbb{R}$ is a Gaussian Mixture Model (GMM) with three components. Means are drawn from $U(0.0, 1.0)^2$ and re-sampled if they fall within $0.01$ of the domain boundary or outside the domain. Covariances are initialized diagonally with log-uniform entries in $\exp(U(\log 0.0001, \log 0.0005))$, and then randomly rotated to obtain full covariance matrices. Component weights follow $\exp(N(0,1)) + 1$, normalized across components, to provide meaningful weight to each component.

Expert meshes are constructed by refining a uniform coarse mesh with element volume $0.01$ using a threshold-based heuristic that accounts for the load function and gradient jumps across element facets [77, 78, 52]. The local error indicator for element $M_i$ is given by

$$\mathrm{err}(M_i) = h_i^2 \|q\|_{L^2(M_i)}^2 + h_i \|[\![\nabla u \cdot \mathbf{n}]\!]\|_{L^2(\partial M_i)}^2, \tag{10}$$

where $h_i$ denotes the characteristic length of $M_i$, and $[\![\nabla u \cdot \mathbf{n}]\!]$ denotes the jump in the normal derivative of $u$ across facets of $M_i$, where $\mathbf{n}$ is the outward unit normal. This estimator highlights regions with strong source terms or large inter-element gradient discontinuities. Elements are marked for refinement if $\mathrm{err}(M_i) > \theta \cdot \max_j \mathrm{err}(M_j)$ with $\theta = 0.85$ fixed across all data points. Marked elements are refined via a conforming red-green-blue scheme [79], followed by Laplacian smoothing after each refinement step.

Each data point comprises a random domain, source term, and corresponding expert mesh. Additionally, we vary task difficulty by controlling the number of refinement steps. We use $25$ steps for an *easy* variant, $50$ steps for *medium*, and $100$ for *hard*. We solve the equation on each intermediate mesh and extract the solution per vertex as a vertex-level input feature to our MPN. In addition, we use the evaluation of $q$ at each vertex as a node feature. We use analogous features evaluated at pixel positions for the image baselines.

## C.2 `Laplace`

The `Laplace` dataset emulates heat conduction or fluid transport through lattice structures during, e.g., compression-based manufacturing processes. It follows the same setup and refinement procedure as `Poisson` (cf. Appendix C.1), but solves the Laplace equation

$$\int_\Omega \nabla u \cdot \nabla v \, \mathrm{d}\boldsymbol{x} = 0 \quad \forall v \in V.$$

We impose a complex Dirichlet boundary condition based on a GMM, applied only to the inner boundary (i.e., the boundaries of the holes). The GMM has means sampled from $U(0.1, 0.9)^2$ and covariances with diagonal entries drawn from $\exp(U(\log 0.005, \log 0.01))$, followed by random rotation. The domain consists of a parameterized family of lattice-like geometries. Each instance contains a uniform grid of $k \times k$ square holes, with $k \in [5, 10]$ and hole sizes drawn from $U(0.04, 0.075)$. Holes are placed evenly, ensuring uniform ligament thickness throughout the lattice.

The refinement procedure is identical to that used in `Poisson`. Since there is no load function, i.e., $q = 0$ for the Laplace equation, the local error indicator simplifies to

$$\text{err}(M_i) = h_i \left\| \llbracket \nabla u \cdot \mathbf{n} \rrbracket \right\|^2_{L^2(\partial M_i)}. \tag{11}$$

We use a fixed number of 100 refinement steps for all data points, corresponding to the *hard* setup of `Poisson`. Each data point consists of a domain, boundary condition, and expert mesh. We solve the equation on each intermediate mesh and use the solution at each vertex as an input feature to our MPN.

### C.3  `Airfoil`

We sample airfoil geometries from the UIUC AIRFOIL COORDINATES DATABASE[4], each with a randomly selected angle of attack. Meshing is performed using GMSH-AIRFOIL-2D[5], which utilizes a task-specific heuristic to generate high-quality meshes with large inflow/outflow regions and fine resolution near the airfoil. We generate 30 meshes, each placing the airfoil at the center of a circular domain within $[0, 1]^2$. The mesh size is set to 0.01 near the airfoil and 0.25 at the outer boundary, yielding approximately 20 000 vertices per mesh.

### C.4  `Beam`

Beam geometries are widely used in mechanical engineering to study structural responses under load, for example in the context of non-linear elasticity [80, 81]. We generate adaptive beam geometries in GMSH [25]. We start with elongated rectangular domains, sampled from height $h$ and length $l$

$$h \sim \mathcal{N}(0.5, 0.05), \quad l \sim \mathcal{N}(10.0, 1.0).$$

Randomly placed disks are subtracted from the domain. The $i$-th disk has radius

$$r_i \sim \mathcal{U}(0.25h, 2.0h),$$

and its center is placed at

$$x_i \sim \mathcal{U}(x_{i-1} + 1.5r_i, x_{i-1} + 20.0r_i), \quad y_i \sim \mathcal{U}(0, h),$$

using an initial reference position $x_0 = 0.1l$ to sample the first disk. Disk placement proceeds sequentially until the beam end is reached. A minimum part thickness of 0.001 is enforced. Meshing uses a manually crafted and carefully tuned heuristic that ensures fine resolution near disks and in thin regions of the geometry.

### C.5  `Console`

`Console` uses data obtained from a real-world scenario in the automotive industry. We have a parameterized family of 3D geometries representing a car's seat crossmembers. The geometries are obtained using ONSHAPE[6] and feature various sharp bends as well as up to two circular holes. Tetrahedral meshes for this dataset are generated by a human expert using ANSA[7]. The expert is initially presented with a coarse mesh, on which they iteratively select regions to refine, specifying the target element size of each selected region. The resulting meshes are optimized for downstream strength and durability analyses, but our experiments are conducted solely on the meshes and their underlying geometry.

### C.6  `Mold`

Injection molding is a key process for manufacturing thin, complex components in high-volume industrial settings [82, 83]. We select plane-like geometries from the ABC: A BIG CAD MODEL dataset [84], aligning them such that the longest dimension lies along $x$ and the shortest along $z$. This standardization does not affect the rotation-invariant *AMBER*, but helps the *Image* baselines.

---

[4]`https://m-selig.ae.illinois.edu/ads/coord_database.html`

[5]`https://github.com/cfsengineering/GMSH-Airfoil-2D/tree/main`

[6]`https://www.onshape.com/`

[7]`https://www.beta-cae.com/ansa.htm`

Geometries are normalized so that the longest in-plane dimension is 1, and their thickness is rescaled to $z \sim \mathcal{U}(0.06, 0.09)$. Each geometry is duplicated three times with varying injection point locations, which are provided as process conditions and influence the meshing strategy. Geometries are imported into ABAQUS [85] and manually meshed by an expert using the standard tetrahedral meshing algorithm. Meshes are tailored for injection molding, with 4–6 elements across thickness and local refinement at holes, edges, and injection points. Mesh generation takes approximately 20 minutes per geometry, depending on complexity.

## D    Training Setup and Hyperparameters

### D.1    Hardware and Compute

All graph-based methods are trained on an NVIDIA 3090 GPU. The image-based methods are instead trained on an NVIDIA A100 GPU to accommodate the memory requirement of the comparatively high-resolution images. Each method is given a computational budget of up to 36 hours, although most methods, including *AMBER*, usually converge after 4-12 hours, depending on the considered dataset. We train every method for five seeds. We evaluate four methods on eight datasets, counting Poisson (*easy/medium/hard*) separately, and four additional methods on three datasets. We additionally have a total of 31 additional experiments across three datasets. Combined, this yields an estimated total compute of $8[\text{hours}] \times 5[\text{seeds}] \times (8 \times 4 + 4 \times 3 + 31) = 3000[\text{hours}]$. A comparable amount was used for preliminary runs and hyperparameter tuning.

### D.2    Training

We implement all neural networks in PyTorch [86] and optimize using ADAM [87]. We use a learning rate of $1.0e$-3 and a linear learning rate scheduler with a warmup from 0 to the full learning rate during the first $10\,\%$ of training. We apply weight decay of $1.0e$-6. We train for a total of $25\,600$ mini-batches for Poisson and Laplace, Airfoil, and $51\,200$ mini-batches for Beam, Console and Mold.

### D.3    Node and Edge Features

In addition to the dataset-specific features, as described in Appendix C each node is assigned features for the average sizing field of adjacent elements, as provided in Equation 3, and the vertex degree. As edge features, we use the Euclidean distance between vertex positions and an approximate curvature, defined as the signed angle between the averaged surface normals of the edge's endpoints. The curvature lies in $[-1, 1]$, with positive values for convex and negative values for concave regions. Since all features are invariant to Euclidean transformations, the architecture is invariant to rotation, translation, reflection, and vertex permutation [21].

### D.4    *AMBER* Hyperparameters

The MPN of *AMBER* consists of 20 separate message passing steps for all datasets. Each message passing step uses separate two-layer MLPs and LeakyReLU activations for its node and edge updates. We apply Layer Normalization [88] and Residual Connections [89] independently after each node and edge feature update, and use Edge Dropout [90] of $0.1$ during training. The final node features are fed into a two-layer MLP decoder. All MLPs use a latent dimension of $64$. We experimented with slightly different parameterizations in preliminary experiments, finding that *AMBER* is relatively insensitive to the details of the underlying MPN. We provide an overview of *AMBER* hyperparameters in Table 4.

## E    Mesh Generation

We use GMSH [25] for mesh generation. For simplicity, we clip the predicted sizing fields during mesh generation to $(0.8 \min\{f_e(M_i^*)\}, 1.25 \max\{f_e(M_i^*)\})$, with $M_i^* \subseteq M^*$, $M^* \in \mathcal{D}$, i.e., to a range around the most extreme values seen during training. Here, $f_e$ is the element-wise sizing field introduced in Section 3.2. This is only done during mesh generation, and does not impact the model predictions or the loss of Equation 2. We further constrain the mesh generation process of *AMBER* to

Table 4: *AMBER* parameters and experiment configuration (variable names as used in the main text)

| Section | Parameter | Variable | Value |
|---------|-----------|----------|-------|
| Optimization | Optimizer | | ADAM |
| | Learning rate | | $1.0 \times 10^{-3}$ |
| | Learning rate scheduler | | linear with $10\%$ warm-up |
| | Weight decay | | $1.0 \times 10^{-6}$ |
| MPN | Aggregation function | $\bigoplus$ | mean |
| | MPN steps | $L$ | 20 |
| | Activation function | | Leaky ReLU |
| | Edge dropout | | 0.1 |
| | MLP layers | | 2 |
| | Latent dimension | | 64 |
| AMBER | Refinement steps | $T$ | 3 |
| | Maximum buffer size | | 500 meshes |
| | Buffer addition frequency | $k$ | 8 samples every 128 batches |
| | Training steps | | 25 600 or 51 200 (task-dependent) |
| | Batch size | | 500 000 graph nodes plus edges |
| | Sizing field scaling | $c_t$ | $1.618^{T-t-1}$ |

a budget of $1.5 \max\{|M_i^*|, M_i^* \subseteq M^*, M^* \in \mathcal{D}\}$ elements, i.e., to $150\%$ of the mesh elements of the largest mesh in the training dataset. To ensure that this budget is met, we employ a conservative heuristic that estimates the number of elements in a newly generated mesh from a given sizing field, and then computes a scaling factor such that the new mesh does not exceed the available number of elements. This constraint makes training more predictable by preventing very large meshes and thus unexpected peaks in runtime between training epochs. While this constraint is also active during inference, we find that it practically never activates after the training has converged.

# F   Metrics

## F.1   Density-Aware Chamfer Distance (DCD)

We evaluate mesh similarity using the DCD [73], a symmetric, exponentiated variant of the Chamfer distance that accounts for multiple points in one set matching a single point in the other. Given vertex sets $\mathcal{V}_1$ and $\mathcal{V}_2$, the DCD is defined as

$$d_{\mathrm{DCD}}(\mathcal{V}_1, \mathcal{V}_2) = \frac{1}{2}\left[ \frac{1}{|\mathcal{V}_1|} \sum_{v \in \mathcal{V}_1} \left(1 - \frac{1}{n_v} e^{-\|\mathbf{p}(v) - \mathbf{p}(\hat{v}(v, \mathcal{V}_2))\|_2}\right) \right.$$
$$\left. + \frac{1}{|\mathcal{V}_2|} \sum_{v \in \mathcal{V}_2} \left(1 - \frac{1}{n_v} e^{-\|\mathbf{p}(v) - \mathbf{p}(\hat{v}(v, \mathcal{V}_1))\|_2}\right) \right], \tag{12}$$

where $\hat{v}(v, \mathcal{V}') = \arg\min_{v' \in \mathcal{V}'} \|\mathbf{p}(v) - \mathbf{p}(v')\|_2$ is the nearest neighbor, and $n_v$ is the number of points in the other set for which $v$ is the nearest neighbor. The DCD is a purely geometric metric that treats both vertex sets as samples from an unknown density over the domain.

## F.2   $L^2$ Error

We additionally evaluate mesh similarity using a symmetric relative projected $L^2$ error between the vertex-based sizing fields of the evaluated and expert meshes. This metric complements the purely geometric DCD by quantifying discrepancies in local element sizes. Let $f$ and $f^*$ denote the vertex-based sizing fields of Equation 3 on the evaluated mesh $M$ and expert mesh $M^*$, respectively. We use the interpolant $\mathcal{I}$ from Equation 1 to evaluate each sizing field at the vertex positions of the opposite mesh. The symmetric relative projected $L^2$ error is then defined as

$$d_{\mathrm{L2}}(M, M^*) = \frac{1}{2}\left( \frac{\left\|f^*(v_j^*) - \mathcal{I}_{M^*}(f)(\mathbf{p}(v_j^*))\right\|_2}{\left\|\mathcal{I}_{M^*}(f)(\mathbf{p}(v_j^*))\right\|_2} + \frac{\left\|f(v_j) - \mathcal{I}_M(f^*)(\mathbf{p}(v_j))\right\|_2}{\left\|\mathcal{I}_M(f^*)(\mathbf{p}(v_j))\right\|_2} \right), \tag{13}$$

where $\| \cdot \|_2$ denotes the discrete $\ell^2$ norm over vertices.

### F.3   Error Indicator Norm

For `Poisson`, we evaluate *ASMR* and *AMBER* using the norm of the error indicator of Equation 10, i.e.,

$$d_{\mathrm{err}}(M) = \|\mathrm{err}(M_i)\|_2 = \sqrt{\sum_{M_i \in M} \mathrm{err}(M_i)^2}. \tag{14}$$

In contrast to the above metrics, the error indicator norm approximates the remaining simulation error for a given mesh, independent of some reference mesh or vertex set. It naturally decreases for finer meshes, but quantifies how well a given mesh works for downstream simulation for its budget. We thus evaluate the Expert, *ASMR* and *AMBER* for different target mesh granularities on a Pareto front of number of mesh elements compared to this norm.

## G   Baselines and Variants

The following sections provide detailed setups for all baselines and variants used in our experiments. Unless mentioned otherwise, the baseline and variant experiments follow the setup and hyperparameters described in Appendix D.

### G.1   *GraphMesh*

*GraphMesh* [24] uses a two-stage GCN [91] to extract geometric information from polygonal domains. It constructs a local copy of the boundary graph for each coarse mesh vertex, encoding relative features to all boundary vertices. These features are mean value coordinates [74], spatial distances, and mesh-hop counts. Thus, each coarse vertex is represented by an individual boundary graph that contains features of the boundary relative to this vertex. This construction limits *GraphMesh* to polygonal domains, which in our case restricts it to the `Poisson` datasets. These graphs are processed by a single-layer GCN, and the resulting embeddings are pooled to yield one latent vector per coarse vertex of the original mesh. To enable load-specific sizing field prediction, the same vertex-level features used in *AMBER* are appended to these embeddings. For the `Poisson` datasets, these features include vertex degree, interpolated sizing field, load function value, and solution value at the vertex position. The combined features are used as node inputs to a second GCN stage consisting of 6 residual graph convolutional layers with 128 dimensional hidden states. *GraphMesh* is trained using a Mean Average Error to the target sizing field and does not apply normalization. We find that *GraphMesh* quickly starts to overfit, especially on `Poisson` (*easy*), likely due to poor generalization capabilities of its GCN and the construction of the geometry embedding. To compensate, we reduce the number of training steps to $3\,200/6\,400/12\,800$ for `Poisson` (*easy/medium/hard*). We tune the resolution of the underlying mesh by dataset for optimal validation performance.

In *GraphMesh (Variant)*, we instead apply the loss in the inverse-softplus space, as in Equation 2, and add input/output normalization. We also use 20 layers with dimension 64 to match *AMBER*'s MPN.

### G.2   *Image* Baseline

The *Image* baseline [22] operates on discretized domain images. In 2D, we use 512 pixels along the longest axis. We evaluate other resolutions in Appendix H.8. In 3D, we use 96 voxels along the longest axis. We follow the original setup and use a U-Net [92] with 64 initial channels and 5 down- and up-convolution blocks. Each block contains 2 convolutions with kernel size 3, followed by batch normalization and a ReLU activation. After each down-convolution, we apply max-pooling with kernel size and stride 2 to halve the resolution and double the number of channels. The up-convolutions reverse this process, and skip connections are added between corresponding depths. We use 2D and 3D convolutions, batch normalization and pooling operations for the 2D and 3D datasets, respectively. For task-specific datasets, i.e., `Poisson` and `Laplace`, we generate a uniform background mesh with roughly one element per pixel and compute an FEM solution on this mesh to yield our input features. For `Poisson`, we additionally include the load function evaluated at each pixel. Finally, for all datasets, we add a binary mask that indicates if a given pixel or voxel is inside

the domain as an input feature. We also mask the loss accordingly, only predicting and training on pixels within the domain. The *Image* baseline is trained on a regular MSE loss over pixel-wise predicted and target sizing fields.

The *Image (Variant)* extends the *Image* baseline to the loss of Equation 2 and input/output normalization. We evaluate both choices individually in Appendix H.8.

### G.3  *AMBER (1-Step)*

We experiment with a variant of *AMBER* that only uses a single mesh generation step, i.e., that predicts vertex-level sizing field on a uniform mesh, and uses this to generate the adaptive mesh. This variant explores *AMBER* without the ability to generate and act on an intermediate meshes, i.e., on a fixed sampling resolution for the predicted sizing field. We keep all *AMBER* hyperparameters the same, but omit all parts of the method that depend on iterative mesh generation. Since *AMBER (1-Step)* heavily depends on the resolution of its input mesh, we tune this resolution separately for each dataset for optimal validation performance.

### G.4  Adaptive Swarm Mesh Refinement (*ASMR*)

For *ASMR* [31, 51] we integrate the `Poisson` dataset into the provided codebase[8], replacing the original mesh generator with GMSH and adapting the dataset parameters to match Appendix C.1. We also adapt the batching scheme to that used for *AMBER* to prevent too-large batches, sampling from the RL replay buffer until the combined number of graph nodes and edges reaches $500\,000$.

We adopt the MPN architecture and training schedule proposed by *ASMR*, using 2 MPN steps. Preliminary experiments with more message passing steps showed no improvement, which is consistent with prior observations on RL model scaling [93, 94].

We use the reward function proposed by *ASMR* [31]. Given an element $M_i^t$ at refinement step $t$, the reward is

$$\mathbf{r}(M_i^t) = \frac{1}{V(M_i^t)} \left( \mathrm{err}(M_i^t) - \sum_j \mathbf{Q}_{ij}^t \mathrm{err}(M_j^{t+1}) \right) - \alpha \left( \sum_j \mathbb{I}(M_j^{t+1} \subseteq M_i^t) - 1 \right), \quad (15)$$

where $\alpha$ is a scalar that controls the trade-off between accuracy and mesh complexity, and is given to the policy as context, and $\mathbf{Q}$ maps refined elements to their parents. The local error $\mathrm{err}(M_i^t)$ is computed by integrating the element-wise solution against a reference solution on a high-resolution mesh. This reference mesh is obtained by uniformly refining the initial mesh six times. Further refinement was found to be computationally infeasible.

The reward function includes a $1/V(M_i^t)$ scaling term. In *ASMR*, both reward and evaluation are based on integration against a fine reference mesh, making the scaling consistent with the objective. We adopt the same reward and optimization, limiting *ASMR* to 6 uniform refinement steps. Appendix H.9 explores a variant that replaces the integrated error with the error indicator, allowing deeper refinement. In both cases, we evaluate using the error indicator, as a sufficiently fine uniform reference mesh is infeasible for our datasets. Under this metric, the scaling biases refinement toward small elements and can lead to a gap to expert performance. In preliminary experiments, we tried to remove the scaling term, which led to unstable training and non-convergence.

We apply an adaptive element penalty during training by sampling $\alpha$ from a predefined range that yields mesh sizes comparable to `Poisson` (*easy/medium/hard*). At inference time, we evaluate across a range of 20 geometrically spaced $\alpha$ values, producing meshes of varying resolution and corresponding indicator error for a full comparison.

## H  Extended Results

### H.1  $L^2$ Error Evaluations

Throughout our experiments, we primarily assess supervised approaches using the Density-Aware Chamfer Distance (DCD) to the expert mesh. Here, we complement this with evaluations based on

---

[8] `https://github.com/niklasfreymuth/asmr`

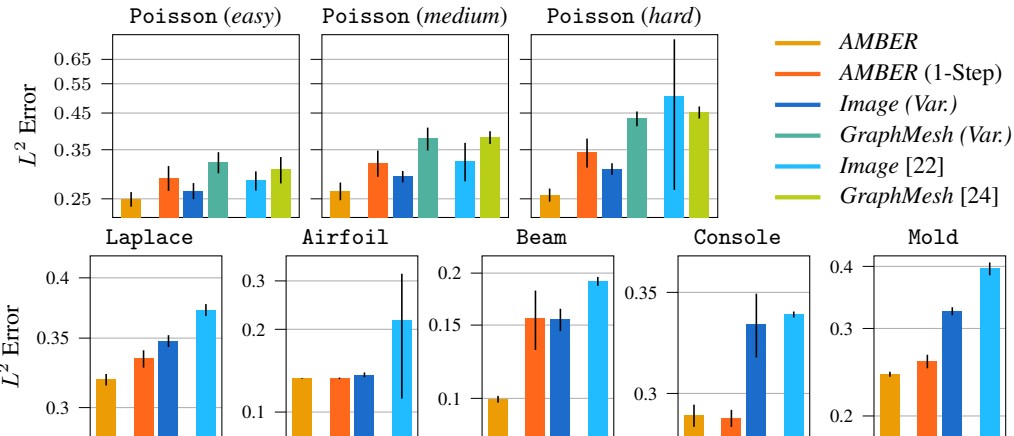

Figure 7: $L^2$ error across datasets and supervised methods. Overall trends are consistent with Figure 3. *AMBER* shows larger relative improvements on datasets like `Poisson` and `Beam` compared to baselines. On `Console`, *AMBER (1-Step)* slightly outperforms *AMBER*, but with overlapping error bounds.

the $L^2$ error defined in Appendix F.2. Figure 7 reports $L^2$ errors across all datasets and supervised methods. While scales are different across datasets, the general trends closely mirror those in Figure 3, with only minor differences in relative performance. On the $L^2$ error, *AMBER* outperforms published baselines on all datasets, and shows a slightly larger advantage over the variants compared to the DCD on, e.g., `Poisson` and `Beam`. For `Console`, *AMBER (1-Step)* performs well on the $L^2$ metric, slightly improving over *AMBER*, although error bounds overlap.

## H.2 Error Indicator Evaluations

We evaluate the norm of the error indicator of Equation 14 for `Poisson` (*hard*) and `Laplace`, i.e., for tasks that use a concrete underlying system of equations. Table 5 shows this error indicator norm and the number of used mesh elements, to account for the norm naturally decreasing with higher element budgets. We find that *AMBER* closely adheres to the element budget of the expert heuristic that was used to generate the data, and that it matches the expert in terms of error indicator. In contrast, many other supervised methods either fail to produce meshes with similar numbers of elements, or have worse error indicator norms, suggesting poor refinements and sub-optimal downstream simulation. These trends highlight *AMBER*'s utility for downstream simulations and validate the use of DCD as a proxy for downstream simulation error.

Table 5: Error indicator norm for `Poisson` (*hard*) and `Laplace` for the expert heuristic and different supervised methods. Overall trends are consistent with Figure 3, validating the use of DCD as a proxy for downstream simulation error.

| Method | Poisson | | Laplace | |
|---|---|---|---|---|
| | Err. Norm | #Elements | Err. Norm | #Elements |
| *AMBER* | $0.031 \pm 0.001$ | $27859.7 \pm 1583.1$ | $2.555 \pm 0.050$ | $27622.5 \pm 943.1$ |
| *AMBER* (1-Step) | $0.032 \pm 0.001$ | $28780.9 \pm 2196.8$ | $2.568 \pm 0.039$ | $27488.7 \pm 706.0$ |
| Image (Var.) | $0.034 \pm 0.001$ | $24836.3 \pm 1213.0$ | $2.697 \pm 0.062$ | $26745.2 \pm 866.7$ |
| Image | $0.082 \pm 0.071$ | $130571.3 \pm 119228.3$ | $3.235 \pm 0.174$ | $29297.8 \pm 8065.7$ |
| GraphMesh (Var.) | $0.042 \pm 0.007$ | $46841.7 \pm 15014.0$ | – | – |
| GraphMesh | $0.034 \pm 0.001$ | $31378.2 \pm 4776.3$ | – | – |
| Expert | $0.033$ | $25625.2$ | $2.766$ | $25130.5$ |

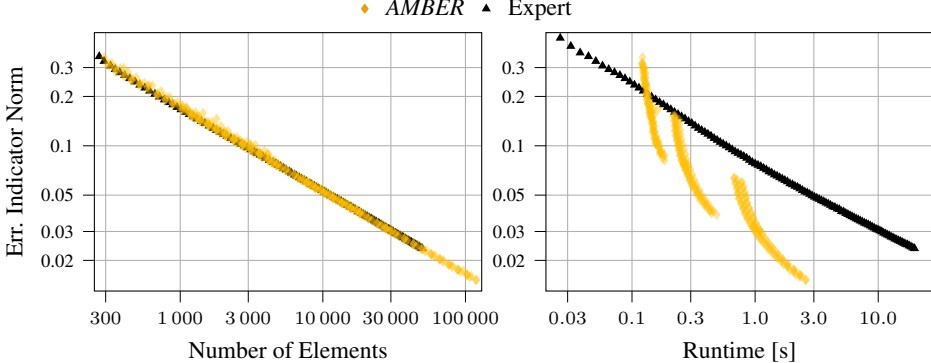

Figure 8: Log-log plot of error indicator norm versus number of mesh elements (**left**) and runtime (**right**) for *AMBER* and the expert across `Poisson` (*easy/medium/hard*). Lower left is better. Each marker shows the mean over the test set for a given seed and target mesh resolution. **Left**: As in Figure 4, *AMBER* achieves comparable error to the expert heuristic for a given element budget. **Right**: For a given training dataset, i.e., any of `Poisson` (*easy/medium/hard*), *AMBER* produces roughly the same intermediate meshes, only adapting to the element budget via the scaling constant $c_T \in [0.5, 2.0]$ at the last step. This process causes a distinct runtime curve for each training dataset. *AMBER* scales better with the element budget than the expert heuristic, eventually achieving a speed-up of more than $10\times$ for meshes with more than $30\,000$ elements.

Table 6: Runtime breakdown of `Poisson` (*easy/hard*) in milliseconds. Mesh generation is the most expensive step, and becomes more costly as the number of mesh elements increases.

| Category | Poisson (*easy*) | | Poisson (*hard*) | |
|---|---|---|---|---|
| | Mean runtime (ms) | % of total | Mean runtime (ms) | % of total |
| Mesh to graph conversion | 15.815 | 8.91 | 94.620 | 8.37 |
| Adding hierarchical graph | 11.219 | 6.32 | 12.606 | 1.11 |
| Model forward | 59.963 | 33.80 | 155.915 | 13.79 |
| Mesh generation | 90.406 | 50.96 | 867.760 | 76.73 |

### H.3   Runtime Experiments

We explore *AMBER*'s runtime behavior across different mesh granularities by training on `Poisson` (*easy/medium/hard*) datasets. We evaluate each trained model by setting the last step's scaling constant $c_T \in [0.5, 2.0]$, as also done in Figure 4. Figure 8 compares the error indicator norm against both the number of mesh elements and the total runtime for *AMBER* and the expert heuristic. Since the scaling constant only comes into effect at the last mesh generation step, the training dataset significantly influences runtime, with distinct curves for models trained with `Poisson` (*easy/medium/hard*). *AMBER* attains an error comparable to the expert heuristic for all element budgets. However, *AMBER* scales significantly better with larger numbers of elements. For large meshes, *AMBER* eventually outperforms the heuristic by more than an order of magnitude, taking less than 3 seconds to generate a mesh with more than $100\,000$ elements. We similarly find that *AMBER* takes less than 5 seconds to accurately imitate a 3D mesh on both `Console` and `Mold`, where a human expert needs roughly 15 to 20 minutes for refinement.

Considering the cost of the individual components of *AMBER*, Table 6 shows that, for $c_T = 1$, mesh generation quickly dominates runtime, taking up more than $50\,\%$ of total cost for `Poisson` (*easy*) and jumping to more than $75\,\%$ for `Poisson` (*hard*). This relative increase in cost is explained by the $O(N \log N)$ scaling of the mesh generation step, which outscales the linear graph-related operations, including the MPN forward, especially for finer meshes. Notably, *AMBER* acts on coarse intermediate meshes, and that the expensive last generation step is also required for the one-step baselines.

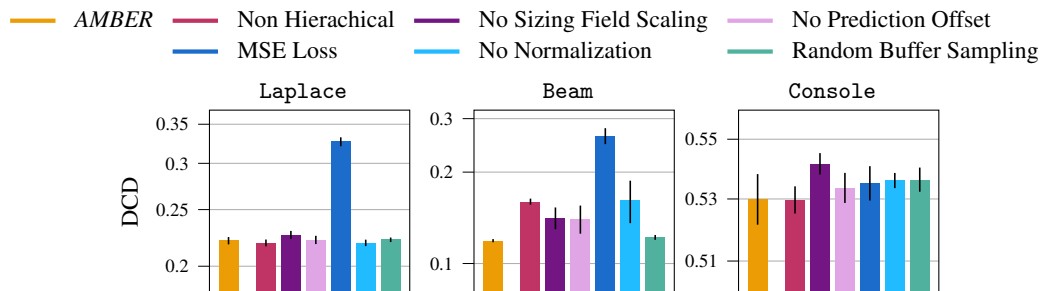

Figure 9: Ablation study on *AMBER* using Density-Aware Chamfer Distance (DCD) across three datasets. Each bar represents a variant of the model with one component removed or modified. Using a non-hierarchical MPN, omitting the prediction offset, or sampling newly generated meshes randomly degrades performance moderately, depending on the task. Omitting normalization or using a regular MSE loss leads to substantially worse generated meshes.

### H.4 Algorithm Design

We evaluate the importance of several of *AMBER*'s components on `Laplace`, `Beam` and `Console`. To evaluate the impact of the loss of Equation 2, we compare against an *AMBER (MSE Loss)* variant using a direct MSE loss between the softplus-transformed predictions and the sizing field targets, i.e.,

$$\mathcal{L} = \frac{1}{|V|} \sum_{j=1}^{V} (\log(1 + e^{x_j}) - y_j)^2. \tag{16}$$

For the algorithmic components, we first replace the stratified sampling for the replay buffer with uniform sampling over all intermediate meshes (*AMBER (Random Buffer Sampling)*). This results in an over-representation of meshes with many prior refinements, leading to a skewed and unbalanced training distribution. Next, we disable the hierarchical mesh representation, feeding only the non-hierarchical graph $\mathcal{G}$ into the MPN (*AMBER (Non-hierarchical)*). This reduces consistency in the receptive field across and within meshes, as regions with higher local resolution require more message passing steps. We also ablate the normalization (*AMBER (No Normalization)*) and the offset term $b_j$ of Section 3.3 (*AMBER (No Prediction Offset)*). Finally, we remove the scaling of sizing fields for intermediate meshes by setting the refinement constant of Section 3.3 to $c_t=1$ for all $t$ (*AMBER (No Sizing Field Scaling)*). While this does not directly impact optimization, it significantly increases intermediate mesh sizes, slowing down mesh generation during training inference, and reducing the number of meshes that fit in a training batch. Figure 9 presents the results of aforementioned algorithmic variants. We find that *AMBER* consistently performs on par with or better than its variations across all datasets. Replacing our loss with a regular MSE leads to the largest degradation in performance, consistently yielding worse meshes than *AMBER* across datasets. Depending on the dataset, different algorithmic components have different impact. The hierarchical graph representation is crucial on `Beam`, as it requires long-range message passing to capture the spatial dependencies of the elongated geometry. The softplus-transformed loss is essential for `Laplace` given its high element scale variation. The sizing field scaling only improves mesh quality slightly, but decreases the size of intermediate meshes, speeding up training and inference. Other factors like normalization and buffer sampling have smaller effects, but still generally yield modest benefits.

### H.5 Sizing Field Parameterization

We experiment with different parameterizations of the predicted sizing field on `Laplace`, `Beam` and `Console`. Given an expert mesh $M^*$, we consider using the vertex-interpolated expert sizing field $f(v_j)$, as defined in Equation 3 to define labels $y_j = \mathcal{I}_{M^*}(f)(\mathbf{p}(v_j))$ using the interpolant of Equation H.5. We call this variant *AMBER (Interpolated Labels)*.

Additionally, we consider a version that predicts a piecewise-constant sizing field $\hat{f}_e(M_i)$ on the elements $M_i$ instead of a piecewise-linear sizing field $\hat{f}(v_j)$ on the vertices $v_j$. Here, the corresponding interpolant $\mathcal{I}_M(f_e)$ is just the union over the element's predictions evaluated at their subdomain, i.e.,

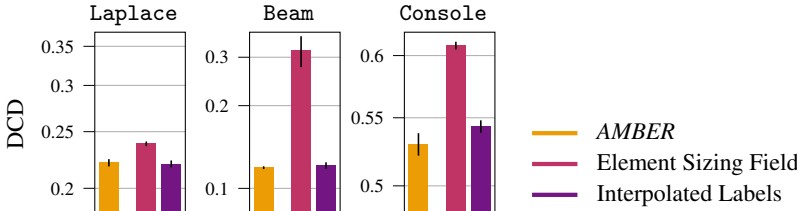

Figure 10: DCD comparison across tasks for different sizing field parameterizations. Interpolating the labels closely matches *AMBER*'s parameterization, reflecting similar optimization objectives. In contrast, using a piecewise-constant sizing field on the elements yields worse meshes on `Beam` and `Console`, likely due to reduced expressiveness.

$$\mathcal{I}_M(f_e)(\mathbf{z}) = \begin{cases} f_e(M_i), & \text{if } \mathbf{z} \in \Omega_i \text{ for some } i, \\ f_e(M_{i'}), & \text{otherwise, where } i' = \arg\min_i \|\mathbf{z} - \mathbf{p}(M_i)\|, \end{cases}$$

where $\mathbf{p}(M_i)$ denotes the position of the element's midpoint. We assign each element the integrated sizing field of all expert elements that it contains, i.e., we compute a volume-weighted average of the sizing field values from the fine mesh elements whose midpoints lie within the coarse element. Let $f_e^*(M_k^*)$ be the sizing field on the fine elements and $V(M_k^*)$ their volume. For each element $M_i$ of the current mesh, we compute the target sizing field as

$$y_i = \begin{cases} \displaystyle\sum_{k \in \mathcal{J}_i} \frac{V(M_k^*)}{\sum_{k \in \mathcal{J}_i} V(M_k^*)} f_e(M_k^*), & \text{if } \mathcal{J}_i \neq \emptyset, \\ f_e(M_k^*), & \text{if } \mathbf{p}(M_i) \in \Omega_k^* \text{ for some } M_k^*, \\ f_e(M_{k'}^*), & \text{otherwise, where } k' = \arg\min_k \|\mathbf{p}(M_i) - \mathbf{p}(M_k^*)\|, \end{cases}$$

where $\mathcal{J}_i = \{j \mid \mathbf{p}(M_k^*) \in M_i\}$ is the set of expert elements whose midpoints lie within the element $M_i$. If there are no such elements, we first attempt to find an expert element $M_{j'}$ that contains the midpoint of $M_i$. If that also fails, the meshes represent different discretizations of the underlying domain. Here, we fall back to nearest-neighbor interpolation using the element midpoint positions. We adapt the MPN input accordingly, constructing the graph $\mathcal{G}$ over mesh elements and element neighborhood. We use the same graph node and edge features, except for the neighborhood size, and always evaluate position-dependent features at the element midpoint. This process yields a variant *AMBER (Element Sizing Field)*.

Figure 10 visualizes results. We find that *AMBER (Interpolated Labels)* performs very similarly to *AMBER*, likely because both optimize a similar objective. While there are small differences in the concrete sizing field targets, especially for early generation steps and coarser input meshes, both parameterizations work well. Here, both parameterizations provide targets that aim to coarsen too-fine regions, while increasing the resolution in too-coarse regions of the current mesh, eventually converging to very similar generated meshes. In contrast, *AMBER Element Sizing Field* predicts a piecewise-constant sizing field over mesh elements. While this works well on `Laplace`, the reduced expressiveness of this parameterization compared to the piecewise-linear interpolant of Equation 1 yields significantly worse meshes on both `Beam` and `Console`.

## H.6 Data Efficiency

Figure 11 assesses *AMBER*'s data efficiency. All other training settings are held constant, and evaluation is performed on the original test set. Accurate mesh generation is achieved with as few as five training meshes and corresponding geometries. Using more samples consistently improves performance. On `Laplace`, where training data can be easily generated via the expert heuristic, there are additional improvements for 100 instead of 20 meshes.

*AMBER*'s efficient use of data likely stems from the local, per-node loss in Equation 2 and the symmetry-preserving features and structure of the MPN architecture.

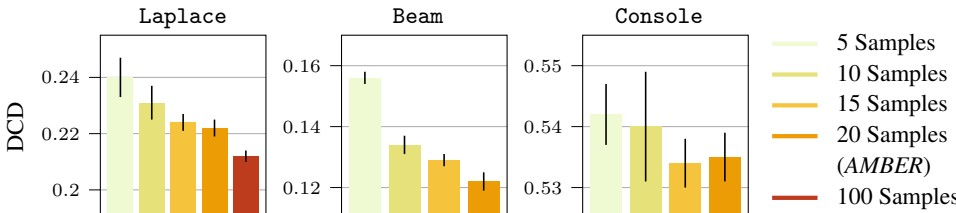

Figure 11: DCD comparison for *AMBER* with different numbers of training samples for `Laplace`, `Beam` and `Console`. *AMBER* performs well with as little as 5 samples, but steadily improves for up to 20 samples. On `Laplace`, where additional training data is easy to generate, there is a moderate improvement for 100 instead of 20 train meshes and geometries.

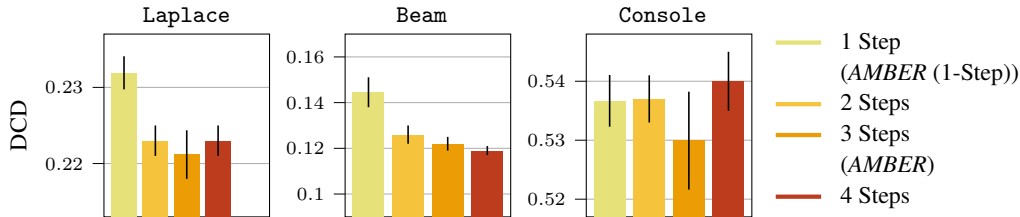

Figure 12: DCD comparison for *AMBER* with different numbers of mesh generation steps for `Laplace`, `Beam` and `Console`. *AMBER* improves for more mesh generation steps, and converges at around three steps. A single mesh generation step is insufficient for accurate generations, likely because it acts on a fixed mesh resolution. Results for *AMBER* and *AMBER* (1-Step) are taken from Figure 3.

### H.7    Mesh Generation Steps

We evaluate how *AMBER* behaves for different numbers of mesh generation steps. In particular, we use three generation steps in all main experiments, and have a single step for *AMBER* (1-Step) as a baseline that acts on a tuned but fixed mesh resolution per task. Figure 12 shows that *AMBER* improves for more mesh generation steps, converging at roughly three steps. Despite tuning the initial mesh size, a single mesh generation step is insufficient for optimal performance, presumably because it does not allow for arbitrarily fine sizing field resolution. In contrast, starting from two mesh generation steps, *AMBER* learns to predict the sizing field used to generate its intermediate meshes, allowing for a flexible, adaptive sizing field representations.

### H.8    Image Ablations

We explore the behavior of the *Image (Variant)* baseline on the `Laplace` task. We vary image resolution and remove either input/output normalization (*Image (No Normalization)*) or the loss from Equation 2, replacing the latter with the MSE loss over the pixel-wise sizing fields (*Image (MSE Loss)*). Omitting both components recovers the original *Image* baseline. In all cases, we still use a softplus to transform the predictions, as we find that directly predicting a sizing field leads to worse performance and unstable mesh generation. Figure 13 shows that performance improves with image resolution, although gains diminish at finer scales. Since the *Image (Variant)* enforces a uniform resolution by design, adapting to high-detail regions becomes prohibitively expensive, leading to substantial waste in less sensitive areas. In contrast, *AMBER* allows for variable sampling resolutions of the predicted sizing field by design, ensuring a more efficient prediction process, especially for highly adaptive meshes. Other than that, both normalization and our loss are crucial for accurate mesh generation, which is consistent with the *AMBER* ablations in Section H.4.

### H.9    *ASMR* (Error Indicator)

We experiment with a version of *ASMR* that uses the error indicator in its reward function, i.e., sets $\text{err}(M_i^t)$ in Equation 15 to Equation 10, leaving the rest of the reward unchanged. To further adapt the resulting *ASMR (Error Indicator)* version to our setup, we disable normalization of the

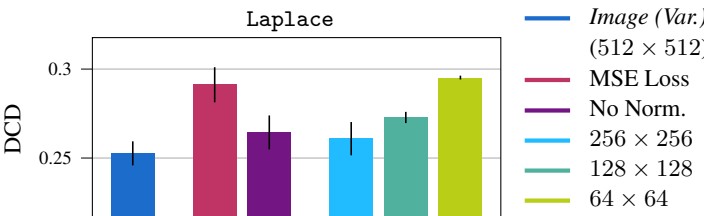

Figure 13: DCD comparison for different *Image (Variant)* ablations. Both the loss of Equation 2 and normalization are crucial for *Image (Variant)*. Performance improves with higher image resolutions, although the rate of improvement eventually slows down.

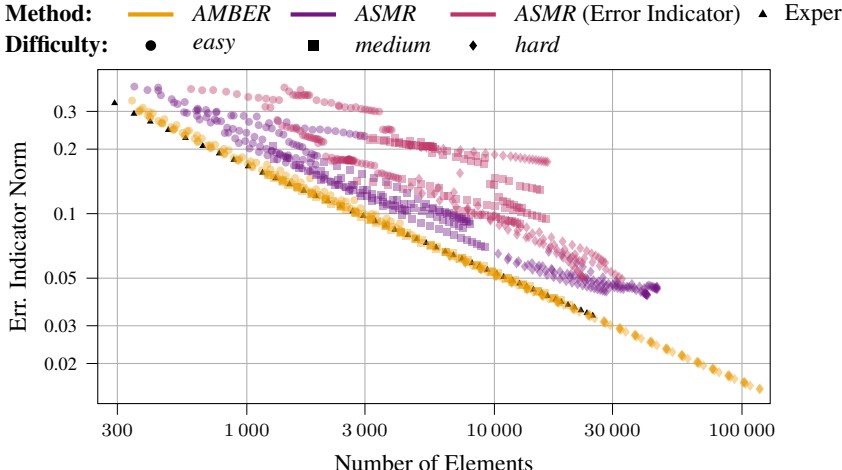

Figure 14: Log-log plot of error indicator norm versus number of mesh elements (lower left is better) for *AMBER*, *ASMR*, *ASMR (Error Indicator)* and the expert across `Poisson` (*easy*, *medium*, *hard*). Each marker shows the mean over the test set for a given seed and target mesh resolution. This figure overlays Figure 4 with *ASMR (Error Indicator)*. We find that training *ASMR* on the indicator error yields less reliable, more noisy policies. However, as this *ASMR* variant is no longer constrained to a fixed refinement depth, it does not degrade as strongly for high-resolution meshes.

initial errors, as we found this to be unstable when using the indicator, and adapt the MPN architecture to 10 message passing steps. We then increase the number of refinement steps to $7/9/11$ for `Poisson` (*easy/medium/hard*), allowing for elements with maximum refinement depth to be of the same size as the minimum expert elements.

Figure 14 overlays Figure 4 with *ASMR (Error Indicator)*. We find that this method performs worse than *ASMR*, presumably because the error indicator is less expressive than the integrated reward. In comparison to the integrated reward, the indicator is noiser, yielding low relative contrast for elements of the same scale. This imbalance makes the reward function harder to optimize, reducing the consistency of the resulting policy. Yet, the indicator does not constraint the refinement depth, allowing for higher mesh resolutions and thus less saturation in simulation quality for finer meshes.

# I  Visualizations

We provide additional visualizations for *AMBER* on all datasets, and for all methods on `Poisson` (*hard*). We visualize the first test data point on the first trained seed for all methods. All visualizations include the expert mesh for reference, and zoom in on a representative region of the geometry.

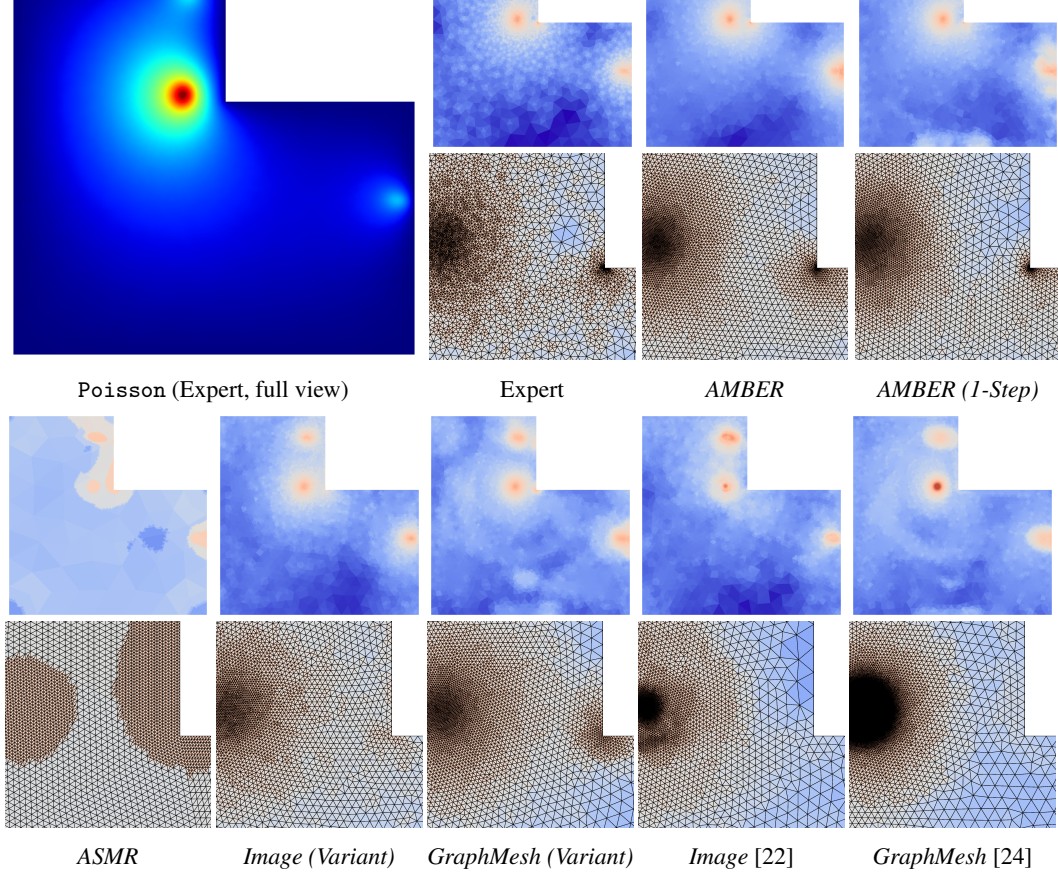

Figure 15: Expert mesh and generated meshes for all baseline methods and *AMBER* on the Poisson (*hard*) dataset. The enlarged view of the expert mesh shows the FEM solution, while other plots show the element size, with red indicating smaller elements. *AMBER* yields more accurate and adaptive meshes, especially in regions with high resolution variability. *ASMR* has a constrained depth, leading to too-uniform refinements, and both *Image* [22] and *GraphMesh* [24] fail to correctly estimate sizing fields in local regions.

## I.1 Baseline Comparisons

Figure 15 visualizes mesh outputs from all baseline methods and *AMBER* on the Poisson (*hard*) dataset. *AMBER* produces more accurate and adaptive meshes, particularly in regions requiring fine detail and large variation in local mesh resolution. In contrast, the baseline methods exhibit artifacts or provide overly smooth or uniform sizing fields. For example, *ASMR* is constrained by the depth of its reference mesh, leading to too-uniform meshes, while *GraphMesh* [24] and the *Image* [22] baseline greatly over- and under-estimate local regions.

## I.2 Full Rollouts

Figures 16 and 17 illustrate full *AMBER* rollouts, showing the iterative mesh generation process from $t=0$ to $t=T=3$ across all datasets. Figure 16 also visualizes the FEM solution on the expert for reference. Across datasets, each generation step incrementally refines the mesh, adding geometric detail and improving alignment with the target solution. The refinement constant $c_t$ introduced in Section 3.3 ensures that early iterations produce coarser meshes, reducing computational cost in the initial stages.

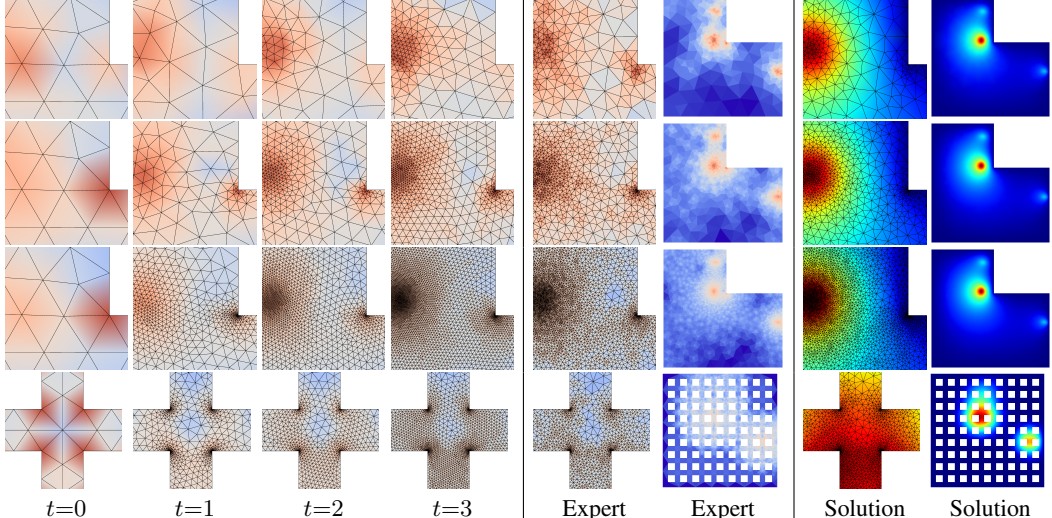

Figure 16: Close-ups of *AMBER* rollouts on the Poisson (*easy/medium/hard*) and Laplace datasets from $t{=}0$ to $t{=}T{=}3$. Left, middle: For $t{<}3$, the colorscale denotes the prediction, otherwise the element size. Right: The colorscale shows the FEM solution on the zoomed-in and full domain for the expert. Successive *AMBER* steps produce increasingly refined meshes that better match the expert, improving sampling resolution for the next sizing field prediction. The refinement constant $c_t$ from Section 3.3 controls mesh granularity over time, enabling coarse and efficient early steps.

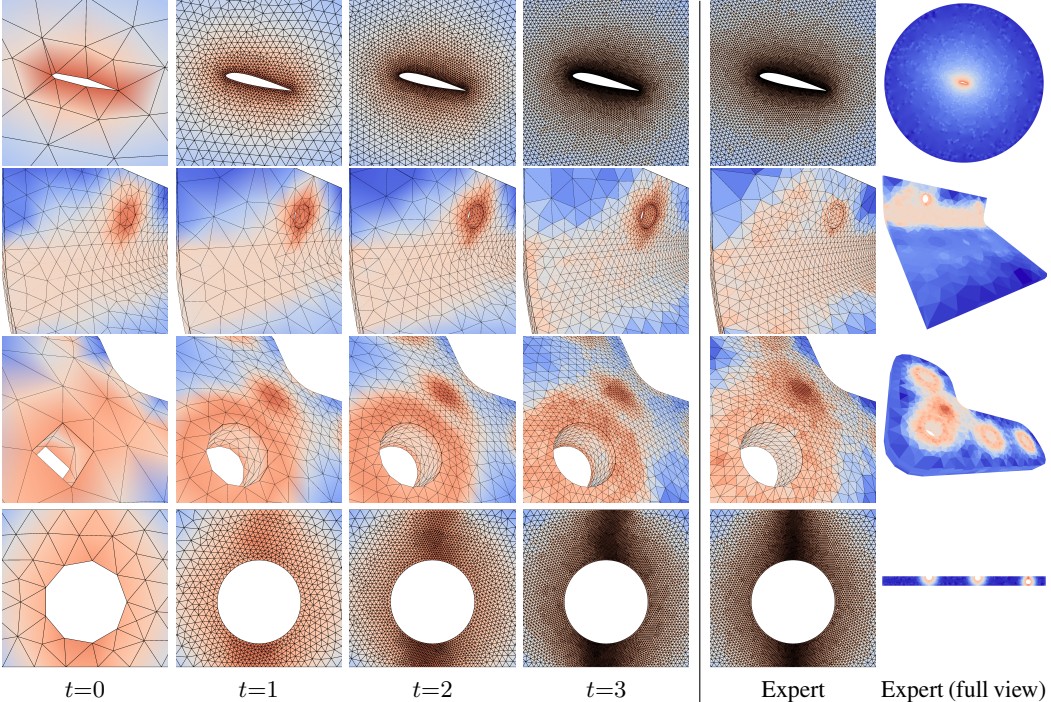

Figure 17: *AMBER* rollouts across the Airfoil, Console, Mold and Beam. The colorscale denotes predictions for $t{<}3$, and element size otherwise, with red indicating smaller values. As in Figure 16, each step provides an increasingly detailed mesh, improving the next prediction's sampling resolution.

