# OpenReview forum: "AMBER: Adaptive Mesh Generation by Iterative Mesh Resolution Prediction"
_NeurIPS.cc/2025/Conference — NeurIPS 2025 poster_

### Official Review · Reviewer_h6nt · 2025-06-29

**Clarity:** 3
**Significance:** 3
**Originality:** 3
**Rating:** 4
**Confidence:** 5

**Summary:**

This paper introduces AMBER - Adaptive Meshing By Expert Reconstruction an Adaptive Mesh Generation (AMG) algorithm used to generate meshes for finite element methods for solving stationary PDEs. The problem setting is formulated as a supervised learning problem, where the target is the element sizing field (ESF) extracted from expert meshes and a MPN style hierarchical GNN is trained to predict the ESF over the PDE domain. Secondarily the ESF is input into the existing frontal Delauney algorithm implemented in GMSH for generating the adapted meshes.

The literature review in the paper correctly sights existing works in this line of research that predict mesh sizing field, however they point several advances on the existing works. The two main novelties of this work are the iterative, instead of 1-shot, mesh generation pipeline and the contribution of the 6 new expert mesh datasets.

A key component of the iterative pipeline is the experience replay buffer, a trick adopted from the RL baseline ASMR, that facilitates training at different lengths scales by bootstrapping a dataset preventing distribution shift from coarse to refined meshes. The bootstrapped dataset is created via projecting the expert mesh sizing fields (targets) onto the intermediate meshes generated from training predictions.

Several other training tricks including stratified batching, normalisation, residual prediction, clipping and softplus transform are used to create a stable training pipeline.

The method is evaluated on 6 new meshing datasets derived from expert methods using density aware chamfer distance (DCD) and error indicators as evaluation metrics.

**Questions:**

As stated above I have a few clarifications needed on the iterative procedure andconcerns over the time complexity of the method.
How does the mesh generator consume the per vertex predicted sizing field? I am used to specifying regions or functions for sizing field in GMSH not a pointwise sizing field - providing more details here would be useful for the reader.
How are the GNN parameters of different scales of the GNN message treated? Are they shared or are there different parameters for different length scales?

**Ethical Concerns:**

["NO or VERY MINOR ethics concerns only"]

**Final Justification:**

My previous concerns over evaluation were too strong given the method is PDE independent. The authors have also addressed my concern on time complexity and somewhat on dataset generalisation.

**Limitations:**

Yes

**Quality:**

2

**Strengths And Weaknesses:**

ML based adaptive meshing via prediction of the element sizing field is a technique already employed by multiple works as referenced by the authors. However the iterative mesh generation, in both training and inference, seems novel and key in achieving successful performance of the model. Other strengths of the paper include the detailed explanation of the tricks used in training and the controllability of the resolution of the model. The new datasets provided by the authors include complex tasks, non-convex domains and 3D examples and are a valuable assets for the community.

Clarity - I found there is some ambiguity in the explanation of the iterative procedure section 3.1, I had to infer extra understanding from the paragraphs on "Residual Prediction" and "Scaling Sizing Fields". I am still not clear how the iterative procedure exactly works. Is the intermediate mesh refined as in old nodes are kept and new nodes are added. Or at each step is the intermediate used to generate the sizing field but the mesh is thrown away? Then the number of nodes is roughly decided by the c_t parameter? If so what is an example schedule of the {c_t} and how does this typically translate into number of nodes?

Also in training based on the "residual prediction" it is unclear how the "current discrete sizing field"  is generated. Is it the expert sizing field or the previous step prediction?

Evaluation - Whilst the evaluation is done using the DCD distance metric against the expert meshes I think this misses the point that adaptive meshing is intended to reduce FEM approximation error which is not reported. All baseline M2N/UM2N/ASMR have some actual FE error based metric. Without seeing this evaluation it is difficult to know if the generated meshes actually perform better in terms of error reduction and solution stability.

Timing - Also no meshing times are provided. This is concerning because adaptive meshing is intended to accelerate FE methods by optimising the trade off between error and complexity. I am confident the method scales in the size of the mesh as evidenced by the use of reasonably large meshes (up to 100k nodes) in the experiments. However I have some concerns on the time complexity of the approach. There are a few reasons the method might be prohibitively slow. At each step in the iteration, I understand, a full new mesh is generated, rather than iterative refinement. I also notice at each step there is a requirement to; i) construct the hierarchical graph using multiple meshes via nearest neighbours, ii) calculate surface normal edge features, iii) add labels of the ESF for the replay buffer, all these step may incur significant cost.

One fundamental requirements for scaling this aproach is that the framework requires expert meshes. Currently the datasets are quite small (20 examples). The authors show evidence AMBER generalize to higher number of elements (100k versus 30k seen in training) and on the Lapalce 2D lattice problem with few-shot learning (but still in-distribution) lattices. The question remains does the framework generalise to a new tasks? Or would it need expert meshes and retraining?

My concerns over evaluation and timing are the main factor in my currently low score.

---

> ### Author Rebuttal · Authors · 2025-07-27
>
> We thank the reviewer for their thorough review, and especially their critical assessment of the clarity of writing for our iterative mesh generation procedure. In the following, we will clarify the mentioned points and address the individual concerns raised by the reviewer.
>
> > AMBER [is] used to generate meshes for finite element methods for solving stationary PDEs. […] the target is the element sizing field (ESF)
> >
>
> We would like to clarify that AMBER may generate meshes independent of an underlying PDE. In our experiments, four datasets rely solely on geometry, and while the two PDE-specific datasets are stationary, the method is not limited to stationary cases and datasets like ```Airfoil``` and ```Console``` are created with non-stationary simulations in mind.
>
> While the sizing field describes the element sizes, it is defined on vertices rather than elements, yielding a piecewise‑linear rather than piecewise‑constant field.  We detail and justify this choice in Appendix H.3.
>
> > […] at each step is the intermediate used to generate the sizing field but the mesh is thrown away? Then the number of nodes is roughly decided by the c_t parameter? If so what is an example schedule of the {c_t} and how does this typically translate into number of nodes?
> >
>
> AMBER generates a new mesh at each step from predicted sizing fields, using previous meshes only as sampling points to predict the sizing field on. We briefly contrast adaptive mesh refinement methods in the “Related Work” and will further clarify this relationship in the revision.
>
> The $c_t$ parameter is a scalar applied to the predicted sizing field. As mentioned in Table 3 in Appendix D.4, we set it to the golden ratio $1.618^{T-t-1}$ for all experiments, with $c_{T-1}=1$ leading to no scaling for the last generation step. The mesh size scales roughly in $\frac{1}{c_t}^d$, where $d$ is the dimension of the data.
>
> > Also in training based on the "residual prediction" it is unclear how the "current discrete sizing field" is generated. Is it the expert sizing field or the previous step prediction?
> >
>
> The current discrete sizing field is generated by projecting the characteristic edge lengths of the elements of the *current* mesh onto its vertices, as described by Equation 3. We use neither the expert sizing field nor the previous prediction directly. We will clarify this relationship in the revised version of this paper.
>
> > I think this misses the point that adaptive meshing is intended to reduce FEM approximation error which is not reported. All baseline M2N/UM2N/ASMR have some actual FE error based metric.
> >
>
> We fully agree that adaptive meshing is intended to reduce FEM approximation error. Most of our datasets are designed to evaluate general‑purpose mesh generation without a specific associated PDE. We assume optimal expert meshes and therefore expect that accurate imitation leads to near‑optimal meshes for downstream simulations.
>
> However, we do have a concrete PDE for both ```Laplace``` and ```Poisson``` and already evaluate the FEM error indicator norm for the latter in Figure 4. The table below reports mean and two times standard error over five seeds for both datasets for AMBER, the baselines, and the expert heuristic. We provide DCD and element counts for reference.
>
> | ```Laplace``` | Err. Indicator Norm | DCD | #Elements |
> | --- | --- | --- | --- |
> | Amber | 2.555 ± 0.050 | 0.222 ± 0.003 | 27622.5 ± 943.1 |
> | Amber (1-Step) | 2.568 ± 0.039 | 0.232 ± 0.002 | 27488.7 ± 706.0 |
> | Image (Var.) | 2.697 ± 0.062 | 0.253 ± 0.006 | 26745.2 ± 866.7 |
> | Image | 3.235 ± 0.174 | 0.328 ± 0.027 | 29297.8 ± 8065.7 |
> | Expert | 2.766 | N/a | 25130.5 |
>
> | ```Poisson``` (*hard*) | Err. Indicator Norm | DCD | #Elements |
> | --- | --- | --- | --- |
> | Amber | 0.031 ± 0.001 | 0.224 ± 0.004 | 27859.7 ± 1583.1 |
> | Amber (1-Step) | 0.032 ± 0.001 | 0.243 ± 0.011 | 28780.9 ± 2196.8 |
> | Image (Var.) | 0.034 ± 0.001 | 0.239 ± 0.010 | 24836.3 ± 1213.0 |
> | Image | 0.082 ± 0.071 | 0.454 ± 0.167 | 130571.3 ± 119228.3 |
> | GraphMesh (Var.) | 0.042 ± 0.007 | 0.330 ± 0.032 | 46841.7 ± 15014.0 |
> | GraphMesh | 0.034 ± 0.001 | 0.265 ± 0.005 | 31378.2 ± 4776.3 |
> | Expert | 0.033 | N/a | 25625.2 |
>
> AMBER achieves lower error indicators than the baseline methods while keeping in the expert’s element budget. The strong correlation between the error metric and DCD across methods further supports the use of DCD as a reliable proxy on datasets where direct FEM error is not available. As AMBER performs best in terms of DCD on all six datasets, its meshes would likely perform well on downstream simulations.  Appendix I further supports these findings by showing that AMBER meshes are qualitatively similar to the expert.
>
> Both M2N and UM2N are mesh movement methods that act on a fixed number of mesh elements and require PDE parameters as input. We thus do not consider these methods as baselines. We will make this distinction more explicit in our "Related Work" section in the revision.
>
> > […] no meshing times are provided. […] There are a few reasons the method might be prohibitively slow. […] there is a requirement to; i) construct the hierarchical graph using multiple meshes via nearest neighbours, ii) calculate surface normal edge features, iii) add labels of the ESF for the replay buffer, all these step may incur significant cost.
> >
>
> We decompose AMBER’s runtime into graph construction, hierarchical extension, model forward, and mesh generation. We note that label calculation is only performed during training and has negligible runtime impact. The table below shows how runtimes and relative costs differ between mesh sizes for ```Poisson``` (*easy*/*hard*).
>
> |  | Poisson (easy) |  | Poisson (hard) |  |
> | --- | --- | --- | --- | --- |
> | Category | Mean runtime (ms) | % of total | Mean runtime (ms) | % of total |
> | Mesh to graph conversion | 15.815 | 8.91 | 94.620 | 8.37 |
> | Adding hierarchical graph | 11.219 | 6.32 | 12.606 | 1.11 |
> | Model forward | 59.963 | 33.80 | 155.915 | 13.79 |
> | Mesh generation | 90.406 | 50.96 | 867.760 | 76.73 |
>
> The mesh generation step is in O(N log N) and dominates runtime compared to the linear graph-related operations, especially for larger meshes. We note that AMBER acts on coarse intermediate meshes, and that the expensive last generation step is also required for the one-step baselines.
>
> For reference, we also provide the average total mesh generation times of AMBER and the expert heuristic for  ``Poisson``` (*easy/medium/hard*) in the table below.
>
> | Difficulty | Method | Total Time (s) | #Elements | Err. Indicator Norm |
> | --- | --- | --- | --- | --- |
> | Easy | Expert | 0.221 | 1091.1 | 0.160291 |
> |  | Amber | 0.150 | 1114.7 | 0.166470 |
> | Medium | Expert | 0.949 | 4472.9 | 0.079407 |
> |  | Amber | 0.271 | 4704.9 | 0.077271 |
> | Hard | Expert | 8.616 | 26372.8 | 0.032524 |
> |  | Amber | 1.024 | 28136.9 | 0.031236 |
>
> AMBER is slightly faster than the expert for small meshes and shows significantly better scaling as mesh size increases. Similarly, AMBER mesh generation for the 3D tasks takes less than five seconds, while the human experts took 15 to 20 minutes per mesh.
>
> > […] does the framework generalise to a new tasks? Or would it need expert meshes and retraining?
> >
>
> We agree that data efficiency and cross-task generalization are important factors of adaptive mesh generation algorithms.
>
> To test AMBER across datasets, we trained a single model on a combined dataset containing ```Poisson``` (*hard*), ```Laplace```, and ```Airfoil```. We used the same architecture and hyperparameters as in the main experiments and simply concatenated the 20 expert meshes per task into 60 total training meshes. We one-hot encode the task in the node features, and zero out task‑specific features when unavailable. We use a shared replay buffer and do not weight the data. The table below shows the mean and two times standard error of the DCD for this *Mixed* setup.
>
> | Method | Poisson | Laplace | Airfoil |
> | --- | --- | --- | --- |
> | Amber (Mixed) | 0.226 ± 0.011 | 0.222 ± 0.005 | 0.102 ± 0.002 |
> | Amber (Original)  | 0.224 ± 0.004 | 0.222 ± 0.003 | 0.103 ± 0.002 |
>
> The *Mixed* setup performs on par with AMBER trained on individual tasks, indicating that the same model can learn meshing strategies across different datasets and domains. We leave a general-purpose mesh generation model for future work.
>
> > How does the mesh generator consume the per vertex predicted sizing field?
> >
>
> Conceptually, we use the vertex-level sizing fields to create a piecewise linear function via the interpolant of Equation 1. Here, we essentially interpret the sizing field as a linear basis on the vertices and interpolate it onto the full mesh.
>
> Code-wise, we use gmsh’s `.pos` options to write the sizing field onto the vertices of a background mesh using “Scalar Triangle” for and “Scalar Tetrahedron” formats. We then provide this background mesh to the meshing algorithm. We refer to the `def update_mesh()`  function in the supplementary code submission for technical details.
>
> > How are the GNN parameters of different scales of the GNN message treated? Are they shared or are there different parameters for different length scales?
> >
>
> AMBER uses the same MPN to process all meshes. Each message passing layer has its own parameters as described in Section 3, but these parameters are shared across mesh generation steps. Differences in scale and hierarchy are handled purely by encoding distances between vertices as edge features and by one‑hot encoding vertices that belong to the hierarchical mesh.
>
> We want to thank the reviewer again for the detailed questions. We hope that our clarifications and new results, particularly regarding generalization and runtime, address the concerns raised. We will add both to the revised paper. We encourage the reviewer to reach out to us during the discussion if there are further questions.

---

> > ### Comment · Reviewer_h6nt · 2025-08-05
> >
> > Dear authors, thank you for the clarifications, I think my critism on evaluation was too strong given that you highlight the algorithm is PDE indepent, and thank you for providing the error metrics for cases associated with a PDE.
> > Thank you for the clarifications on the calculation of the mesh sizing field, golden ratio and number of nodes.
> > I appreciate the runtime breakdown, being able to see it is the standard mesh generation and not the additional computations that dominates. The context of 5 seconds versus 20 minutes for an expert was a useful grounding.
> > The mixed setup is an interesting result which goes towards general purpose/foundational.
> > I update my score accordingly.

---

> > > ### Author Response · Authors · 2025-08-06
> > >
> > > Dear Reviewer,
> > >
> > > Thank you for your kind and constructive reply. We're glad the clarifications were helpful and appreciate your updated assessment. Your feedback has been very valuable in refining our work and its presentation.

---

### Official Review · Reviewer_MLnH · 2025-07-01

**Clarity:** 3
**Significance:** 3
**Originality:** 2
**Rating:** 5
**Confidence:** 3

**Summary:**

This work introduces AMBER, a novel data-driven method for adaptive mesh generation. AMBER iteratively refines a mesh sizing field, which can subsequently be utilized by standard mesh generators for downstream application. During its training phase, the method is supervised by projecting element sizes from expertly crafted meshes onto intermediate meshes. Additionally, a replay buffer is maintained, containing model-generated meshes to facilitate data augmentation and stabilize the training process. Extensive experiments conducted on both 2D and 3D datasets indicate that AMBER consistently produces meshes of superior quality compared to both supervised learning and reinforcement learning baselines.

**Questions:**

Using a graph neural network makes sense but then why a MPN and not for instance a GAT ? Please elaborate.

From an end-user perspective, controllability is essential: did you consider the possibility of conditioning your approach on a user input such as a picture/mask or a specific mesh annotation to enforce -- for instance -- mesh refinement (or simplification) in a user-defined area?

**Ethical Concerns:**

["NO or VERY MINOR ethics concerns only"]

**Final Justification:**

I appreciate the effort and detailed response provided by the authors in the rebuttal. I believe that my concerns have been well addressed, leaving me with a positive opinion about this work. I update my rating from 4: Borderline accept to 5: Accept.

**Limitations:**

yes

**Paper Formatting Concerns:**

The paper is clear and well written. The supplementary material nicely complement the main paper.

Figures are in general clear and relevant.

One thing: Figure 4 is somewhat hard to read --> consider using different tones of pastel colors to render the difficulty: dark tones could mean difficult while light tones would mean easy. The markers are quite hard to decipher.

**Quality:**

3

**Strengths And Weaknesses:**

### Strengths

The paper is clear, well structured, backed by strong experiment results and ablations. The literature review is also addressed seriously and seems quite comprehensive and up to date.

To the best of my understanding of GNN, the paper is mathematically sound.

The provided code is clean, well structured and well documented.

### Weaknesses

I'm a bit confused as it seems AMBER was introduced last year in this paper: https://arxiv.org/pdf/2406.14161
Is the current paper supposed to be a new version of this paper or should rather it be considered as a revision ?

---

> ### Author Rebuttal · Authors · 2025-07-27
>
> We would like to thank the reviewer for the positive review, and we appreciate the feedback that the paper is well-structured, has strong evaluations and is mathematically sound.
>
> The reviewer raised one weakness, namely an arXiv paper they linked. We appreciate the detailed research of the related work and want to highlight that the mentioned arXiv paper was only presented at a non‑archival workshop. According to the NeurIPS guidelines, specifically the second question in the FAQ section, such workshop versions do not preclude submission to NeurIPS:
>
> > *Q: What if I’ve seen similar work in a NeurIPS/ICML workshop?*
> >
> >
> > ***A:** We allow work that has been submitted to non‑archival workshops to be submitted to NeurIPS. To maintain anonymity, do not mention the workshop paper in your review.*
> >
>
> The present submission substantially extends prior non-archival work and has not appeared in any archival venue. It includes new theoretical results, a different parameterization of the sizing field, two additional baselines and four additional datasets, as well as more thorough evaluation. We kindly ask the reviewer to assess the submission without considering the arXiv paper.
>
> We thank the reviewer for the questions they raised, and will provide detailed answers in the following.
>
> > Using a graph neural network makes sense but then why a MPN and not for instance a GAT?
> >
>
> We thank the reviewer for asking about this comparison. MPNs have been shown to be a more expressive class of GNNs than GATs [1,2] and encompass the function class of several classical solvers [3]. As such, they have been widely used for physical simulations [4, 5] and adaptive mesh refinement [6]. We therefore adopt the MPN architecture for AMBER, as it is a natural and popular choice for processing graphs in physics-based applications.
>
> That said, AMBER predicts a scalar sizing field per vertex, leading to relatively smooth predictions across the graph topology. A GAT with edge features, as used in an ablation in [6], could therefore perform similarly well, and a detailed exploration of alternative GNN architectures would be an interesting avenue for future work.
>
> [1] Veličković, Petar. "Everything is connected: Graph neural networks." *Current Opinion in Structural Biology,* 2023.
>
> [2] Bronstein, Michael M., et al. "Geometric deep learning: Grids, groups, graphs, geodesics, and gauges." *arXiv*, 2021.
>
> [3] Brandstetter, Johannes, et al. “Message passing neural PDE solvers”. *ICLR,* 2022.
>
> [4] Pfaff, Tobias, et al. "Learning mesh-based simulation with graph networks." *ICLR*, 2020.
>
> [5] Linkerhägner, Jonas, et al. "Grounding Graph Network Simulators using Physical Sensor Observations." *ICLR*, 2023.
>
> [6] Freymuth, Niklas, et al. "Swarm reinforcement learning for adaptive mesh refinement." *NeurIPS*, 2023.
>
> > did you consider the possibility of conditioning your approach on a user input such as a picture/mask or a specific mesh annotation to enforce -- for instance -- mesh refinement (or simplification) in a user-defined area?
> >
>
> We thank the reviewer for the question and suggestion. The current work does not consider user inputs during runtime, and instead relies on the quality of the expert meshes that the model is trained on.
>
> However, AMBER works by predicting a series of sizing fields, which are scalar functions on the given domain. To integrate user input, one could simply manipulate this sizing field in user-defined regions, e.g., by interpolating between the user’s preferences and AMBER’s suggested sizing field. We show in Figure 4 that AMBER can generate meshes with different numbers of elements by scaling all predictions with a single scalar. To adapt this to user preferences, one could derive a scaling term for each region of the mesh, or even each mesh vertex, from the user annotations.
>
> > Figure 4 is somewhat hard to read
> >
>
> We would like to thank the reviewer for pointing this out and will improve the figure and the extended Figure 11 in the appendix in the revised version of the paper.
>
> We hope that our clarifications, especially regarding the non-archival workshop paper posted by the reviewer, address the concerns raised. If there are any remaining questions, we encourage the reviewer to reach out to us during the discussion for further clarification.

---

> ### Author Response · Authors · 2025-08-06
>
> Dear Reviewer,
>
> Thank you once again for your thoughtful and constructive feedback.
>
> We hope our responses have addressed your concerns. Given the limited discussion period, we would be grateful if you could briefly confirm whether any issues, in particular regarding the workshop paper you mentioned, remain unresolved. We would welcome the opportunity to address remaining concerns and incorporate the necessary improvements in the revision.
>
> Thank you for your consideration and support.

---

### Official Review · Reviewer_hXSF · 2025-07-03

**Clarity:** 4
**Significance:** 3
**Originality:** 3
**Rating:** 5
**Confidence:** 4

**Summary:**

The paper introduces AMBER (Adaptive Meshing By Expert Reconstruction), a supervised framework for adaptive mesh generation (AMG). Starting from a coarse, uniform mesh, a message-passing network (MPN) iteratively predicts a vertex-wise sizing field using a hierarchical graph neural network formulation that keeps receptive fields stable across refinements, after which an off-the-shelf generator (Gmsh) refines the mesh accordingly. Each newly produced mesh is fed back as input, allowing the model to adjust its sampling density over successive steps. Training uses "expert” meshes: AMBER projects their element sizes onto every intermediate mesh and stores these labelled rollouts in a replay buffer, mitigating covariate shift in the DAgger style.

Experiments on six new 2-D/3-D datasets show that, against all baselines runnable on each task, AMBER lowers Density-Aware Chamfer Distance (DCD) and projected L2 errors.  On Poisson, it matches – and slightly surpasses – expert meshes on the high-resolution end ($\approx$ 100 k elements) of the Poisson error-indicator curve. On the Laplace dataset, AMBER remains data-efficient, needing as few as five training geometries.

**Questions:**

1) Have you tried pre-training on one dataset and fine-tuning on another, or training a single model on all six datasets?
2) What fraction of wall-clock time is spent inside Gmsh vs. the GNN forward pass?
3) You fix $T=3$ (three mesh-generation steps).  How does performance change for $T=1,2,4$ in both training and inference?

**Ethical Concerns:**

["NO or VERY MINOR ethics concerns only"]

**Final Justification:**

The authors have sufficiently addressed my questions during the rebuttal period, and I maintain my positive score of **Accept**.

**Limitations:**

Yes.

**Paper Formatting Concerns:**

The paper follows the NeurIPS 2025 formatting.

**Quality:**

4

**Strengths And Weaknesses:**

**Strengths**

- **Thorough Empirical Study (Quality and Significance):** The evaluation pipeline includes ablations on loss, hierarchical graph, buffer sampling, sizing-field parameterisation and data size, as well as comparisons to three baselines: Image (supervised, all datasets), GraphMesh (supervised, Poisson only) and ASMR (RL, Poisson only).

- **Clarity:** The manuscript is well-written, results are well illustrated, and diagrams depicting the pipeline are present, which aid with understanding the proposed approach.
- **Well-motivated Problem (Significance):** The problem is well-motivated. Manual AMG is a recognized bottleneck in FEM workflows, and replacing hand-tuned heuristics with learned sizing-field prediction is valuable.
- **Clean and Modular Design (Originality):** The three-step roll-out with a self-bootstrapped replay buffer is conceptually simple yet effectively addresses covariate shift (a weakness of prior one-shot approaches).

**Weaknesses**

- **Isotropic Sizing Field (Significance):** The current formulation cannot express anisotropic element stretching, a common requirement near boundary layers and shock fronts.
- **Baseline Coverage (Significance):**  Because GraphMesh and ASMR run only on Poisson, the cross-dataset comparison relies mainly on the authors’ Image baseline.
- **Physics-level Validation (Significance):** Only Poisson tasks include an FEM error indicator, while the other five datasets are judged purely on geometric distance.

---

> ### Author Rebuttal · Authors · 2025-07-27
>
> We thank the reviewer for their detailed and positive review, especially for highlighting the rigour of our experimental setup. In the following, we will briefly address the individual questions and concerns raised.
>
> > The current formulation cannot express anisotropic element stretching, a common requirement near boundary layers and shock fronts.
> >
>
> We agree that anisotropic meshing is an important property for adaptive mesh generation approaches. While extending AMBER to anisotropic meshes is out of scope for this work, doing so should be relatively straightforward. Instead of using a frontal Delaunay algorithm for meshing, we would need to use an anisotropic mesh generation algorithm, such as BAMG, which is readily implemented in gmsh. AMBER then needs to provide a tensor instead of a scalar field, which can be realised by adapting its output dimension per vertex and making its GNN architecture equivariant, similar to [1]. We leave this as an interesting avenue for future work.
>
> [1] Bekkers, Erik J., et al. "Fast, Expressive SE(n) Equivariant Networks through Weight-Sharing in Position-Orientation Space." *2024*, ICLR
>
> > Because GraphMesh and ASMR run only on Poisson, the cross-dataset comparison relies mainly on the authors’ Image baseline.
> >
>
> We fully agree that broader baseline coverage would strengthen the comparison. However, existing methods are constrained to specific settings. For example, GraphMesh only works on polygonal domains and ASMR ties its refinement to an underlying FEM, so they cannot be applied to the full range of tasks supported by AMBER. To still provide meaningful comparisons, we extended the image baseline to 3D CNNs for 3D tasks, introduced Image (Var.) as a more optimized image-based approach, and included AMBER (1‑Step) as a general-purpose GNN-based one-step mesh generation method. We additionally provide example meshes in Appendix I for a qualitative comparison to other methods and the expert.
>
> > Only Poisson tasks include an FEM error indicator, while the other five datasets are judged purely on geometric distance.
> >
>
> Most of our datasets only consider expert meshes rather than any specific underlying PDE, and therefore no FEM error indicator is available. These meshes reflect practical engineering scenarios, where the same adaptive mesh may serve multiple downstream simulations without being tied to a single scenario. For example, ```Console``` meshes may be used for various stress tests, and ```Airfoil``` for different kinds of fluid flow simulations.
>
> For the two tasks that do have an associated FEM, i.e., ```Laplace``` and ```Poisson```, we report FEM error indicator norms in the table below. We report the (*hard*) version of ```Poisson``` as it is the most challenging, and also provide the number of elements and the Density-Aware Chamfer Distance (DCD) to the expert mesh. These metrics correspond to Figures 4 and 3 of the main paper, respectively. Values are reported as mean and twice the standard error over 5 seeds. For reference, we also include the error indicator norm and element count of the expert heuristic. All statistics are computed on the test set, while Table 2 in Appendix C reports expert statistics on the training set.
>
> | ```Laplace``` | Err. Indicator Norm | DCD | #Elements |
> | --- | --- | --- | --- |
> | Amber | 2.555 ± 0.050 | 0.222 ± 0.003 | 27622.5 ± 943.1 |
> | Amber (1-Step) | 2.568 ± 0.039 | 0.232 ± 0.002 | 27488.7 ± 706.0 |
> | Image (Var.) | 2.697 ± 0.062 | 0.253 ± 0.006 | 26745.2 ± 866.7 |
> | Image | 3.235 ± 0.174 | 0.328 ± 0.027 | 29297.8 ± 8065.7 |
> | Expert | 2.766 | N/a | 25130.5 |
>
> | ```Poisson``` (*hard*) | Err. Indicator Norm | DCD | #Elements |
> | --- | --- | --- | --- |
> | Amber | 0.031 ± 0.001 | 0.224 ± 0.004 | 27859.7 ± 1583.1 |
> | Amber (1-Step) | 0.032 ± 0.001 | 0.243 ± 0.011 | 28780.9 ± 2196.8 |
> | Image (Var.) | 0.034 ± 0.001 | 0.239 ± 0.010 | 24836.3 ± 1213.0 |
> | Image | 0.082 ± 0.071 | 0.454 ± 0.167 | 130571.3 ± 119228.3 |
> | GraphMesh (Var.) | 0.042 ± 0.007 | 0.330 ± 0.032 | 46841.7 ± 15014.0 |
> | GraphMesh | 0.034 ± 0.001 | 0.265 ± 0.005 | 31378.2 ± 4776.3 |
> | Expert | 0.033 | N/a | 25625.2 |
>
> These new results show a high correlation between the error indicator norm and the DCD, suggesting that expert similarity is a good indicator for downstream performance. AMBER achieves a significantly lower FEM error indicator norm than the baselines while remaining close to the expert’s geometry and element budget. AMBER’s strong performance on both FEM-associated datasets, combined with the correlation between error indicator and DCD, indicates that AMBER meshes would likely perform similarly well on different downstream simulations on the other datasets.
>
> > Have you tried pre-training on one dataset and fine-tuning on another, or training a single model on all six datasets?
> >
>
> We ran additional experiments on a dataset combining ```Poisson``` (*hard*) ```Laplace```, and ```Airfoil```. The experiment uses the same model- and hyperparameters as all other AMBER experiments in the paper, and simply concatenates the 20 training meshes of all tasks into a total of 60 expert meshes for training. For the node features, we additionally one-hot encode the task and zero out task-specific features that are not available for some tasks. We use a shared replay buffer and do not weight meshes from the datasets. The table below shows the mean and two times standard error of the DCD for this *Mixed* setup.
>
> | Method | Poisson | Laplace | Airfoil |
> | --- | --- | --- | --- |
> | Amber (Mixed) | 0.226 ± 0.011 | 0.222 ± 0.005 | 0.102 ± 0.002 |
> | Amber (Original)  | 0.224 ± 0.004 | 0.222 ± 0.003 | 0.103 ± 0.002 |
>
> We find that there is no significant difference in performance to training individual models, suggesting that AMBER can straightforwardly be applied to datasets comprising multiple different geometry families and meshing strategies. These results open up interesting avenues for future work, such as training a general-purpose foundation model for mesh generation.
>
> > What fraction of wall-clock time is spent inside Gmsh vs. the GNN forward pass?
> >
>
> Each AMBER iteration consists of a creating a graph, adding this to a hierarchical graph, forwarding the result through the MPN model, and then generating the next mesh from the predicted sizing field. The fraction of time that each step takes depends on the size of the created mesh. We show results for ```Poisson``` (*easy*) and ```Poisson``` (*hard*) in the tables below.
>
> |  | Poisson (easy) |  | Poisson (hard) |  |
> | --- | --- | --- | --- | --- |
> | Category | Mean runtime (ms) | % of total | Mean runtime (ms) | % of total |
> | Mesh to graph conversion | 15.815 | 8.91 | 94.620 | 8.37 |
> | Adding hierarchical graph | 11.219 | 6.32 | 12.606 | 1.11 |
> | Model forward | 59.963 | 33.80 | 155.915 | 13.79 |
> | Mesh generation | 90.406 | 50.96 | 867.760 | 76.73 |
>
> As meshes become bigger, the graph generation and model forward scale linearly with the size of the mesh, while the mesh generation scales in O(N log N). Since AMBER acts on a series of coarse intermediate meshes, the last generation step is by far the most expensive. A similar generation step is also needed for the one-shot baselines.
>
> > How does performance change for T=1,2,4 in both training and inference?
> >
>
> We thank the reviewer for the insightful question. We ran five AMBER with 1-4 mesh generation steps for five seeds on the datasets that we used for the extended results in Appendix G, i.e., ```Laplace```, ```Beam``` and ```Console```. We report the DCD in the table below.
>
> | #Steps | Laplace | Beam | Console |
> | --- | --- | --- | --- |
> | 1 (Amber (1-Step)) | 0.232 ± 0.002 | 0.260 ± 0.058 | 0.561 ± 0.021 |
> | 2 | 0.223 ± 0.002 | 0.126 ± 0.004 | 0.537 ± 0.004 |
> | 3 (Amber) | 0.222 ± 0.003 | 0.122 ± 0.003 | 0.535 ± 0.004 |
> | 4 | 0.223 ± 0.002 | 0.119 ± 0.002 | 0.540 ± 0.005 |
>
> The results indicate that as few as two mesh generation steps are sufficient, although three steps perform slightly better. Adding an additional fourth step does not generally improve performance. Note that the MPN’s prediction is independent of the step, and that a model trained on, e.g., four steps can also be evaluated for only two steps, potentially at slight degradation of performance because of some irrelevant data in the training replay buffer.
>
> We hope that our clarifications and newly provided results address the concerns raised. If there are further questions about the paper, we encourage the reviewer to reach out to us during the discussion for further clarification. We also want to thank the reviewer again for suggesting the above experiments, and will include them and their results in the revised paper.

---

> > ### Comment · Reviewer_hXSF · 2025-08-05
> >
> > Dear Authors,
> >
> > Thank you for the additional experiments and clarifications. The mixed-dataset training, detailed runtime breakdown, and step-count study directly answer my questions, while the new Laplace FEM results address my physics-level-validation concern. The remaining isotropic-sizing limitation is already acknowledged as future work.
> >
> > Therefore, I maintain my "Accept" recommendation and original category scores.

---

> > > ### Author Response · Authors · 2025-08-06
> > >
> > > Dear Reviewer,
> > >
> > > Thank you again for your positive review. We're pleased to hear that the additional results and clarifications addressed your concerns. We appreciate your continued support and recommendation.

---

### Official Review · Reviewer_1QCu · 2025-07-04

**Clarity:** 3
**Significance:** 2
**Originality:** 2
**Rating:** 4
**Confidence:** 3

**Summary:**

Authors propose AMBER, which introduces a hierarchical Message Passing Neural Network to iteratively predict spatially adaptive mesh sizing fields, guided by expert-labeled meshes. During training, each intermediate mesh’s vertices are “labeled” with target sizes derived from an expert-designed mesh for the same geometry. Through this label projection onto intermediate meshes, the model learns to mimic expert meshing strategies at each refinement step. This is an iterative process wherein, in each step, a graph model takes the mesh at that step as input and predicts the mesh element sizing field. This field is then passed on to the existing meshing tool (gmsh) to re-mesh the geometry. The method employs online data augmentation via a replay buffer of self-generated meshes to handle distribution shift.

The authors evaluate AMBER on six new datasets spanning 2D and 3D geometries (e.g. an L-shaped domain for a Poisson PDE, a lattice for Laplace’s equation, an airfoil shape, a 3D car console component, injection molding plates, and perforated beams). Comprehensive benchmarks on diverse 2D/3D datasets demonstrate substantial improvements over supervised and RL baselines in mesh quality and simulation accuracy. AMBER’s meshes are not just closer to the expert in a mathematical sense, but also improve actual simulation accuracy, as shown by the error-vs-element plots where AMBER’s curve nearly overlaps the expert’s and even trends slightly better for very fine meshes.

The authors introduce 6 novel datasets of geometries and corresponding expert-refined meshes. Some of these expert-refined meshes seem to be generated using heuristic rules, while others seem to be generated by a human expert refining the input mesh. It would be helpful to report how the different human experts refining and heuristic mesh quality differs.

**Questions:**

Is it feasible to train one AMBER model across multiple datasets or a very broad distribution of geometries? If the core strategy is truly task-agnostic, it would be interesting to know if the model could learn a more universal meshing strategy given a sufficiently diverse training set (maybe with additional conditioning on the type of problem)

How sensitive is AMBER to the quality and quantity of expert examples?

Could the authors provide more evidence or reasoning on how the improved mesh quality translates to simulation accuracy or efficiency across tasks?

What are the runtime and memory implications of AMBER compared to traditional adaptive meshing? An analysis of how long it takes to generate a mesh with AMBER (for a given number of elements) vs. a heuristic or an expert doing it manually would be very informative.

**Ethical Concerns:**

["NO or VERY MINOR ethics concerns only"]

**Final Justification:**

The authors effectively addressed my primary concerns on cross-dataset generalization, runtime analysis, and sensitivity to refinement steps. A minor limitation remains regarding AMBER's dependence on the quality and consistency of expert meshes, though the authors clearly acknowledge this. Considering the substantial improvements provided in the rebuttal, I have positively updated my score.

**Limitations:**

Yes

**Paper Formatting Concerns:**

All the methods, model architectures, and empirical improvements are described in good detail.

**Quality:**

3

**Strengths And Weaknesses:**

The lack of experiments on cross-dataset generalization is a missed opportunity to demonstrate broader applicability. The current approach trains a separate model for each dataset/task, and there is no unified model or transfer learning demonstrated across different geometry types. While each model generalizes to new geometries within its domain (e.g., new airfoil shapes or new console designs) as shown in the experiments, it is unclear if a single model could handle multiple categories or if knowledge could be shared.

The paper doesn’t explore how the method might perform if the expert examples are suboptimal or inconsistent, or how many expert examples are needed. This limitation means AMBER’s success is tied to the availability of good training data, which could limit cases where it can be applied (e.g., novel applications with no prior meshing solutions available).

Method should directly evaluate simulation accuracy on the more complex 3D tasks and not just the simple Poisson example.

Discuss runtime or memory costs as mesh size becomes larger. The paper could discuss the sensitivity of the method to the initial mesh or the number of refinement steps (fixed or adaptive?) If fixed, choosing too low a value for refinement steps might limit quality, while too high adds cost but the paper does not detail how it is set, except that it matches the “number of refinement steps used during evaluation”. Clarifying such implementation details would improve reproducibility and clarity.

---

> ### Author Rebuttal · Authors · 2025-07-27
>
> We thank the reviewer for their thorough review and for acknowledging the comprehensive evaluation on our datasets. We address the raised questions below.
>
> > It would be helpful to report how the different human experts refining and heuristic mesh quality differs.
> >
>
> We agree on the need to clarify mesh generation and quality. Appendix C details each dataset, and Table 1 summarizes the process. Each dataset uses human or heuristic meshes with strategies tailored to their applications. ```Poisson``` and ```Laplace``` apply an iterative heuristic based on an estimated FEM error. The other tasks have no single governing PDE associated with them, so their meshes are adapted for practical scenarios where one high‑quality mesh serves multiple simulations. For example, ```Console```   considers stress tests and thus refines around inner radii and bends, while ```Airfoil```  refines around the airfoil center. We will update the main paper to clarify this relationship.
>
> > The lack of experiments on cross-dataset generalization is a missed opportunity […] Is it feasible to train one AMBER model across multiple datasets or a very broad distribution of geometries?  […]
> >
>
> We agree with the reviewer that training a single model across multiple datasets is an important experiment.  We have now additionally trained an AMBER model using the combined training data of ```Poisson``` (*hard*), ```Laplace``` and ```Airfoil```. We use the same experimental setup as in the paper. We append a one‑hot task indicator to the vertex features and zero unused dataset‑specific features. We use a shared replay buffer and do not weight the datasets. The markdown table below presents the Density-Aware Chamfer Distance (DCD) to the expert meshes, which is the metric used in Figure 3. We compare this new *Mixed* setup to the previous per-task AMBER and report mean and two times standard error across five seeds.
>
> | Method | Poisson | Laplace | Airfoil |
> | --- | --- | --- | --- |
> | Amber (Mixed) | 0.226 ± 0.011 | 0.222 ± 0.005 | 0.102 ± 0.002 |
> | Amber (Original)  | 0.224 ± 0.004 | 0.222 ± 0.003 | 0.103 ± 0.002 |
>
> We observe no significant difference from training per‑task models, indicating that AMBER applies directly to datasets spanning multiple geometry families and meshing strategies. We plan to explore broader generalization in future work.
>
> > The paper doesn’t explore how the method might perform if the expert examples are suboptimal or inconsistent, or how many expert examples are needed.
> >
>
> AMBER is a supervised approach and thus assumes that the provided meshes are i.i.d. and optimal. It therefore imitates the given data and produces near‑optimal meshes when the data is optimal. Handling imperfect or inconsistent meshes is an interesting direction but beyond this paper’s scope.
>
> Table 10 in Appendix H.4 reports results for more and fewer expert meshes on ```Laplace```. We further evaluate the DCD across five seeds for 5, 10, 15, and 20 training meshes on ```Laplace```, ```Beam``` and ```Console``` in the table below.
>
> | #Samples | Laplace | Beam | Console |
> | --- | --- | --- | --- |
> | 5 | 0.240 ± 0.007 | 0.156 ± 0.002 | 0.542 ± 0.005 |
> | 10 | 0.231 ± 0.006 | 0.134 ± 0.003 | 0.540 ± 0.009 |
> | 15 | 0.224 ± 0.003 | 0.129 ± 0.002 | 0.534 ± 0.004 |
> | 20 (Default) | 0.222 ± 0.003 | 0.122 ± 0.003 | 0.535 ± 0.004 |
>
> AMBER benefits from more data, but yields accurate adaptive meshes from only 5 training meshes, still performing on par with or better than the best baselines of Figure 3, which are trained on 20 meshes.
>
> > Method should directly evaluate simulation accuracy on the more complex 3D tasks and not just the simple Poisson example.
> >
>
> We thank the reviewer for suggesting more FEM evaluations. While most tasks have no concrete PDE associated with them, as mentioned above, we do simulate for ```Laplace``` and ```Poisson```.
>
> The tables below show the error indicator norm of the simulation alongside DCD and element count across five seeds for AMBER and different baselines. We provide expert statistics on the test set for reference.
>
> | ```Laplace``` | Err. Indicator Norm | DCD | #Elements |
> | --- | --- | --- | --- |
> | Amber | 2.555 ± 0.050 | 0.222 ± 0.003 | 27622.5 ± 943.1 |
> | Amber (1-Step) | 2.568 ± 0.039 | 0.232 ± 0.002 | 27488.7 ± 706.0 |
> | Image (Var.) | 2.697 ± 0.062 | 0.253 ± 0.006 | 26745.2 ± 866.7 |
> | Image | 3.235 ± 0.174 | 0.328 ± 0.027 | 29297.8 ± 8065.7 |
> | Expert | 2.766 | N/a | 25130.5 |
>
> | ```Poisson``` (*hard*) | Err. Indicator Norm | DCD | #Elements |
> | --- | --- | --- | --- |
> | Amber | 0.031 ± 0.001 | 0.224 ± 0.004 | 27859.7 ± 1583.1 |
> | Amber (1-Step) | 0.032 ± 0.001 | 0.243 ± 0.011 | 28780.9 ± 2196.8 |
> | Image (Var.) | 0.034 ± 0.001 | 0.239 ± 0.010 | 24836.3 ± 1213.0 |
> | Image | 0.082 ± 0.071 | 0.454 ± 0.167 | 130571.3 ± 119228.3 |
> | GraphMesh (Var.) | 0.042 ± 0.007 | 0.330 ± 0.032 | 46841.7 ± 15014.0 |
> | GraphMesh | 0.034 ± 0.001 | 0.265 ± 0.005 | 31378.2 ± 4776.3 |
> | Expert | 0.033 | N/a | 25625.2 |
>
> The error indicator norm strongly correlates with DCD, which measures expert similarity, across methods and tasks. This indicates that expert similarity is a good proxy for downstream simulation quality. AMBER consistently attains a lower FEM error indicator norm than baselines while closely matching the expert’s geometry and element budget. Its superior performance and the error–DCD correlation suggest AMBER meshes would perform well in downstream simulation.
>
> > Discuss runtime or memory costs as mesh size becomes larger.  […] What are the runtime and memory implications of AMBER compared to traditional adaptive meshing?
> >
>
> AMBER uses an MPN with runtime and memory costs linear in mesh size. Each AMBER step builds a graph, integrates it hierarchically, forwards through the MPN, and generates the next mesh from the predicted sizing field. The mesh generation step scales in O(N log N) and dominates cost. Since AMBER acts on coarse intermediate meshes, most of this cost incurs at the last generation step, which is also needed for all other methods. We report detailed AMBER runtimes for ```Poisson``` (*easy*/*hard*) in the table below to show how runtime costs shift for different mesh sizes.
>
> |  | Poisson (easy) |  | Poisson (hard) |  |
> | --- | --- | --- | --- | --- |
> | Category | Mean runtime (ms) | % of total | Mean runtime (ms) | % of total |
> | Mesh to graph conversion | 15.815 | 8.91 | 94.620 | 8.37 |
> | Adding hierarchical graph | 11.219 | 6.32 | 12.606 | 1.11 |
> | Model forward | 59.963 | 33.80 | 155.915 | 13.79 |
> | Mesh generation | 90.406 | 50.96 | 867.760 | 76.73 |
>
> Graph-related operations require linear memory cost. AMBER’s iterative process acts on coarser intermediate meshes, and the MPN never needs to process the final mesh. During evaluation, intermediate meshes with >100,000 elements fit on a consumer-grade NVIDIA 3090 GPU, allowing for potentially much finer meshes depending on the application.
>
> > The paper could discuss the sensitivity of the method to the initial mesh or the number of refinement steps (fixed or adaptive?)
> >
>
> We thank the review for the suggestion to discuss the sensitivity to the initial mesh. We use 3 mesh generation steps for all experiments for AMBER, and compare to a AMBER (1-step) baseline that uses only one step.
>
> We tested the sensitivity to the initial mesh for AMBER and AMBER (1-Step) on ```Laplace```. The table below shows DCD over five seeds for different maximum volumes of the initial mesh elements. The `*`  designates the resolution chosen in the paper. The resolution of AMBER (1-step) was tuned to be optimal while keeping the initial mesh smaller than the expert mesh.
>
> |  | Max. Initial Volume | DCD |
> | --- | --- | --- |
> | Amber | 0.001* | 0.222 ± 0.003 |
> |  | 0.0003 | 0.218 ± 0.002 |
> |  | 0.0001 | 0.218 ± 0.002 |
> |  | 0.00003 | 0.218 ± 0.003 |
> |  | 0.00001 | 0.222 ± 0.002 |
> | Amber (1-Step) | 0.001 | 0.360 ± 0.003 |
> |  | 0.0003 | 0.288 ± 0.002 |
> |  | 0.0001 | 0.255 ± 0.003 |
> |  | 0.00003* | 0.232 ± 0.002 |
> |  | 0.00001 | 0.229 ± 0.006 |
>
> AMBER is robust to initial mesh size, whereas AMBER (1‑Step) requires tuning.
>
> We additionally ran experiments on two and four mesh generation steps on ```Laplace```, ```Beam``` and ```Console``` across five seeds, and provide the results in the table below.
>
> | #Steps | Laplace | Beam | Console |
> | --- | --- | --- | --- |
> | 1 (Amber (1-Step)) | 0.232 ± 0.002 | 0.260 ± 0.058 | 0.561 ± 0.021 |
> | 2 | 0.223 ± 0.002 | 0.126 ± 0.004 | 0.537 ± 0.004 |
> | 3 (Amber) | 0.222 ± 0.003 | 0.122 ± 0.003 | 0.535 ± 0.004 |
> | 4 | 0.223 ± 0.002 | 0.119 ± 0.002 | 0.540 ± 0.005 |
>
> AMBER performs well with two steps, improves slightly with three, and shows no gain beyond that, though additional steps may help for larger geometries.
>
> > An analysis of how long it takes to generate a mesh with AMBER (for a given number of elements) vs. a heuristic or an expert doing it manually would be very informative.
> >
>
> The time to generate an expert mesh depends on the underlying task and the type of expert. For ```Console``` and ```Mold```, our human experts took 15-20 minutes per mesh. In contrast, AMBER generates a 3D mesh in <5 seconds. The table below additionally measures runtimes for AMBER and the expert heuristic on ```Poisson```.
>
> | Difficulty | Method | Total Time (s) | #Elements | Err. Indicator Norm |
> | --- | --- | --- | --- | --- |
> | Easy | Expert | 0.221 | 1091.1 | 0.160291 |
> |  | Amber | 0.150 | 1114.7 | 0.166470 |
> | Medium | Expert | 0.949 | 4472.9 | 0.079407 |
> |  | Amber | 0.271 | 4704.9 | 0.077271 |
> | Hard | Expert | 8.616 | 26372.8 | 0.032524 |
> |  | Amber | 1.024 | 28136.9 | 0.031236 |
>
> We find that Amber has a relatively high overhead for small meshes, but becomes significantly faster than the iterative expert heuristic as meshes become larger.
>
>
>
> We thank the reviewer for these suggestions and hope that we addressed all concerns raised. We will include the new results in the revised paper and welcome further questions in the discussion phase.

---

> ### Author Response · Authors · 2025-08-06
>
> Dear Reviewer,
>
> Thank you again for your thoughtful and constructive feedback. We truly appreciate the time and effort you have invested in reviewing our work.
>
> We hope our responses have fully addressed your concerns. Given the limited discussion period, we would be very grateful if you could briefly confirm whether our replies resolve the issues you raised. Should any concerns remain or if you have further suggestions, we would welcome the chance to address them and improve the paper accordingly in the revision.
>
> Thank you for your consideration and support.

---

> > ### Comment · Reviewer_1QCu · 2025-08-07
> >
> > Thank you to the authors for addressing my previous concerns and providing additional evaluations. The clarification on cross-dataset generalization, robustness to expert data quality, and runtime analysis improves my assessment. Some minor concerns remain regarding AMBER’s dependence on the quality of expert-provided meshes and broader generalizability across completely novel tasks without retraining. Nonetheless, I have updated my rating positively based on these clarifications.

---

> > > ### Author Response · Authors · 2025-08-07
> > >
> > > We thank the reviewer for their detailed feedback and for acknowledging our clarifications. We appreciate the updated rating and will add a discussion on the remaining concerns in the revised paper, particularly mentioning current limitations regarding expert mesh quality and generalization to novel tasks. We are glad the additional evaluations helped improve the reviewer's assessment.

---

### Note · Authors · 2025-08-12

We thank the reviewers and AC for their constructive engagement. While the initial reviews were already positive, the rebuttal phase allowed us to further strengthen the paper with targeted new experiments and analyses, as summarized below.

* **Cross-dataset generalization**: A single AMBER model trained on ```Poisson``` (_hard_), ```Laplace```, and ```Airfoil``` matches per-task performance, confirming applicability across geometry families and meshing strategies.

* **Downstream accuracy**: FEM error evaluations for Laplace and Poisson show a strong correlation between DCD and FEM error, with AMBER outperforming all baselines while matching expert element budgets.

* **Data efficiency**: AMBER produces accurate meshes from as few as five expert examples and improves consistently with more data.

* **Runtime and scalability**: Runtime breakdowns show linear graph-processing costs and substantial speed-ups over expert heuristics for large meshes.

* **Robustness**: AMBER is insensitive to initial mesh size and performs well with as few as two refinement steps.

We believe that these additions address key reviewer questions and significantly strengthen the paper’s contributions. We are glad for the helpful feedback and are happy to include these changes in the revision.

---

### Decision · Program_Chairs · 2025-09-17

**Decision:**

Accept (poster)

**Comment:**

The paper proposes a graph neural network to efficiently predict/refine the meshes to match expert-created meshes. The main contribution of this work is the network architecture and the datasets that match real-world settings. The reviewers have unanimous support for this paper, valuing its significance in the task and the quality of the paper’s presentation. There were a few clarifying questions on the papers’ details, and they were resolved during the discussion period. The AC does not find reason to overturn the reviewers’ recommendation.